# Automatic snow type classification of Snow Micro Penetrometer profiles with machine learning algorithms

Julia Kaltenborn[1,2,3,4], Amy R. Macfarlane[1], Viviane Clay[2,5], and Martin Schneebeli[1]

[1]WSL Institute for Snow and Avalanche Research SLF, Flüelastrasse 11, 7260 Davos Dorf, Switzerland
[2]Institute of Cognitive Science, University Osnabrück, Wachsbleiche 27, 49090 Osnabrück, Germany
[3]Mila – Quebec AI Institute, 6666 Rue Saint-Urbain, QC H2S 3H1, Montréal, Canada
[4]School of Computer Science, McGill University, 3480 Rue University, QC H3A 2A7, Montréal, Canada
[5]Numenta, 889 Winslow Street, CA 94063, Redwood City, United States

**Correspondence:** Julia Kaltenborn (julia.kaltenborn[at]mail.mcgill.ca), Martin Schneebeli (schneebeli[at]slf.ch)

**Abstract.** Snow-layer segmentation and classification are essential diagnostic tasks for various cryospheric applications. The SnowMicroPen (SMP) measures the snowpack's penetration force at submillimetre intervals in snow depth. The resulting depth-force profile can be parameterized for density and specific surface area. However, no information on traditional snow types is currently extracted automatically. The labeling of snow types is a time-intensive task that requires practice and becomes infeasible for large datasets. Previous work showed that automated segmentation and classification is, in theory, possible but can not be applied to data straight from the field or needs additional time-costly information, such as from classified snow pits. We evaluate how well machine learning models can automatically segment and classify SMP profiles to address this gap. We trained fourteen models, among them semi-supervised models and artificial neural networks (ANNs), on the MOSAiC SMP dataset, an extensive collection of snow profiles on Arctic sea ice. SMP profiles can be successfully segmented and classified into snow classes based solely on the SMP's signal. The model comparison provided in this study enables SMP users to choose a suitable model for their task and dataset. The findings presented will facilitate and accelerate snow type identification through SMP profiles. Anyone can access the tools and models needed to automate snow type identification via the software repository "snowdragon". Overall, snowdragon creates a link between traditional snow classification and high-resolution force-depth profiles. Traditional snow profile observations can be compared to SMP profiles with such a tool.

## 1 Introduction

The cryosphere covers around 10% of our earth and plays a significant role in stabilizing the earth's climate (on Climate Change , IPCC). Snow cover plays a role in optics, heat, and mass balance and is one of the most significant uncertainties in global climate models (Sturm and Massom, 2017; Steger et al., 2013; Douville et al., 1995). Snow layer segmentation and classification put forth knowledge about the atmospheric conditions a snowpack has experienced (Colbeck, 1987; Fierz et al., 2009). This knowledge helps to discern fundamental snow and climate mechanisms in the Arctic and to analyze polar tipping points. Classification of snow types (also referred to as "snow grain type" or "grain type" in the community) is essential to assess the state of our cryosphere. It is thus of interest for polar, cryospheric, and climate change research (Domine et al., 2019;

King et al., 2015; Sturm and Liston, 2021). Snow type is often better reproduced in detailed snow cover models (Vionnet et al., 2012) than their effective physical properties, especially indirectly structural anisotropy (King et al., 2015). This is especially relevant for active and passive microwave sensing, essential to map the arctic snowpack during polar night (Sandells et al., 2023).

Traditionally, snow stratigraphy measurements are made in snow pits. These pits are dug manually into snowpacks, requiring trained operators and a substantial time commitment. To accelerate these measurements, the SnowMicroPen (SMP), a portable high-resolution snow penetrometer, can be used (Johnson and Schneebeli, 1998). They have demonstrated the SMP as a capable tool for rapid snow type classification and layer segmentation. The measurement results are stored in an SMP profile that consists of the penetration force signal of the measurement tip in Newton and the depth signal indicating how far the tip moved. Afterwards, the SMP profiles must be manually labeled by an expert, which requires time and practice.

To address these shortcomings, Machine learning (ML) algorithms could be used to automate the labeling process. Instead of manually labeling each SMP profile, an ML model can be trained on a few labeled profiles and subsequently reproduce the labeling patterns on other profiles. As a consequence, this would (1) immensely accelerate the SMP analysis, (2) enable the analysis of large datasets, and (3) support interdisciplinary scientists who are unfamiliar with snow type categorization.

Such an automatic classification of SMP profiles helps to find layers with shared properties within a large SMP dataset. By reproducing a trained labeling pattern on new profiles with ML, SMP classification is up-scaled. While it is impossible to manually label and analyse a dataset of thousands of SMP profiles, an ML-assisted classification enables us to conduct completely new analyses. Questions like "How does a typical snow layer in the Arctic look?" suddenly move within reach. Statistical analyses of signal and layer types rely on consistent, large, and fully labeled SMP datasets.

Several previous works have addressed automatically classifying snow types with machine-learning algorithms. The nearest neighbour method of Satyawali et al. (2009) was the first model that automated the segmentation and classification of SMP profiles without needing additional snow pit information. To assign a snow type to an unlabeled data point, the method chooses the most frequent class occurring in the neighbourhood of this data point. The neighbourhood contains the most similar points to the unlabeled data point. Their algorithm predicts five different snow types ("New Snow", "Faceted Snow", "Depth Hoar", "Rounded Grains", "Melt-Freeze"), with an accuracy ranging from 0.68 to 0.94. However, this high performance is only achieved by integrating specific and inflexible expert rules. For example, one rule ensures that no "Faceted Snow", "Depth Hoar", or "Rounded Grains" occur between layers of "New Snow", but precisely this happens under certain circumstances, as they point out. Hard-coded rules might improve the performance of one dataset, but they cannot capture all phenomena and will not generalize well to other datasets. The performance results are also limited by the fact that their testing set consists of only three SMP profiles, i.e. it is not clear how representative their results are. In addition, their results can hardly transfer to the real-world setting because they explicitly exclude any mixed snow type layers. Suppose an automatic segmentation and classification algorithm will work with profiles straight from the field. In that case, this algorithm should be able to handle mixed classes and diverse snow phenomena and be thoroughly tested.

Havens et al. (2013) worked with random forests and SVMs to classify SMP profiles. They used previously segmented SMP profiles and classified the snow type of each layer with the help of a random forest model. They build upon their previous

work with single decision trees (Havens et al., 2010). They trained the model on three different snow types ("New Snow", "Rounded Grains", "Faceted Grains"), achieving error rates between 16.4% and 44.4% (depending on the dataset). Notably, Havens et al. (2013) requires profiles that have been manually segmented beforehand. Since this is done manually, this takes a considerable amount of time, raising the question of to what extent the task has been "automated". Only layers larger than 100 mm (sometimes 20 mm) could be considered due to manual segmentation. In the field, particularly for avalanche risk assessment (Lutz et al., 2007), it is important to detect layers only a few millimetres thick. Improving on the work of Havens et al. (2010) would thus include more snow types, thinner layers, and no need for manual segmentation.

More recently, King et al. (2020a) trained Support Vector Machines (SVMs) on SMP force signals and manual density cutter measurement. Both segmentation and classification are conducted automatically. They distinguish three types of snow grains ("Rounded", "Faceted" and "Hoar") and achieve classification accuracies between 0.76 and 0.83. The profiles were collected on Arctic ice in the same region, which means that the profiles might be more homogeneous than in other datasets. In theory, the model's generalisability could be enhanced by training it on additional, broader datasets. Most importantly, the SVM method by King et al. (2020a) relies on additional manual density cutter measurement, time-intensive snow pit measurements that are not always available. Thus, similarly to Havens et al. (2013), more snow types would make the work more applicable in the field and eliminate the necessity of additional manual density cutter measurements. In summary, previous work showed that supervised machine learning algorithms are a promising pathway to automatic snow grain categorization.

While all these works put forward the task of automated SMP analysis, SMP users still lack a method that can be used in practice. Users need a model that fully automates their SMP analysis: (1) without the need of digging a snow pit, (2) picking layers manually, or (3) constructing specific knowledge rules. Furthermore, SMP users need models to deal with SMP profiles from the field. This implies that (4) the profiles have multiple snow types (more than three) and that (5) no layers are excluded. This study aims to provide models that fully automate SMP analysis and can directly be used, addressing all five mentioned needs.

To this end, we implemented fourteen different machine learning (ML) models and compared their performance on the MOSAiC SMP dataset, consisting of 164 labeled profiles (see Fig. 1)(Kaltenborn et al., 2021). We provide the first comparable performance overview of different models classifying and segmenting SMP profiles. Moreover, we used semi-supervised methods and artificial neural networks (ANNs) for SMP classification.

Results show that especially artificial neural networks (ANNs), such as the long short-term memory (LSTM) and Encoder-Decoder, can produce predictions similar to profiles labeled by experts and achieve the best results among all models. However, the choice of the model depends mostly on the individual needs of an SMP user because factors such as explainability, desired sensitivity to rare classes, available time, and computational resources must be considered.

The work presented here is a methodological contribution. We provide insights into which ML algorithms can be used to automatically and consistently classify large SMP datasets. Our findings can be applied to different SMP datasets or similar data. The more fine-grained contributions of this study are:

- Demonstration that SMP profiles straight from the field can be automatically segmented and classified; without manual preparation of the profiles or additional snow-pit data after training on a smaller set of SMP profiles,

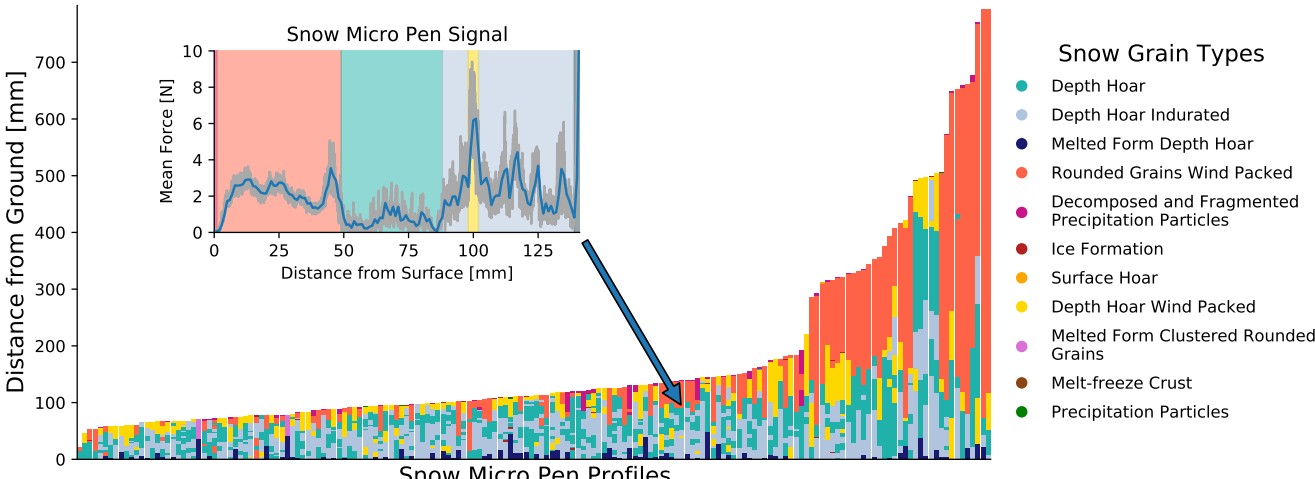

**Figure 1.** All 164 labeled SnowMicroPen (SMP) profiles used for training, validation (80%), and testing (20%). Each bar represents one SMP profile. The colours encode the different snow types. The top of each bar is the air-snow interface and the bottom is the profile's snow-ground interface. The in-picture figure illustrates the force signal (grey) and the mean force signal (blue) of a single SMP profile (S31H0368). The snow-air interface is on the left, and the bottom of the profile is on the right. The background shading in the inset panel and the colors in the main panel represent the labeling of the profiles.

- – Evaluation of semi-supervised models and ANNs for SMP classification,

- – Detailed comparison of different ML models for SMP classification,

– The snowdragon repository which provides the tools to automate SMP labeling.

In the following section (Sect. 2) the data and the classification task and the fourteen different models used in this study are described. In Sect. 3, the models' performances are presented. Subsequently, the results, their limitations, and future work are discussed in section 4. The impact of this work is addressed in the conclusion (Sect. 5). The code and data availability is outlined directly after the conclusion, and a detailed guide on how to use snowdragon with your SMP dataset can be found in 100 Appendix A.

## 2 Methods

### 2.1 Data

All experiments throughout this study use snow data collected during the MOSAiC expedition (October 2019 - September 2020) (Nicolaus et al., 2022). The snow pit measurements conducted include SMP profiles, micro computer tomography 105 (Micro-CT) (Coléou et al., 2001), and near-infrared (NIR) photographs (Matzl and Schneebeli, 2006). Collecting snow profiles

on Arctic sea ice is especially challenging: A) Only a few hours were available to perform all measurements within one snow pit. B) The measurements must be conducted with wind velocities up to 25 m/s and temperatures of $-30°$ Celsius. C) Changing personnel, i.e. different operators were conducting the snow pit measurements. As a result, traditional stratigraphy analysis and in-situ snow grain classification from snow pits carry operator biases. Merkouriadi et al. (2017) could measure only 27 snow pits with stratigraphy under similar conditions.

In contrast, during the MOSAiC expedition, several thousand (3680) SMP profiles were collected. Out of the 269 snowpit events which included SMP measurements, 102 had NIR measurements and 103 had micro-CT profiles collected simultaneously. 71 snowpit events had all three measurements (SMP, NIR and micro-CT). We encountered eleven different snow types. Refer to Fierz et al. (2009) for descriptions of the different snow types referenced here and a classification guideline for snow particles visually observed.[1]

The main measurements collected were signal profiles from the Snow Micro Penetrometer since it provides profiles fast, with little physical labour, and independently from the person who measures them. Of the 3680, 164 profiles from the cold season (January – May 2020) were labeled and evaluated here (see Fig. 1). The labels expressed by color in Fig. 1 indicate which snow type is found at the respective position of the profile. In this study, we focus only on profiles of cold snow, as no standardized interpretation of SMP force profiles exists for wet snow. All profiles collected in the cold season are referred to as "MOSAiC winter data" in the following. Micro-CT and NIR were recorded whenever possible to validate the subsequent labeling of the SMP profiles. More details on the collection methods can be found in Macfarlane et al. (accepted 2023). A figure giving a comparison of these instruments can be seen in Appendix B in Figure B2.

The labeling of the SMP profiles was conducted by two snow experts and is based on the properties of the force signal (magnitude, frequency, and gradient) and the signature of the SMP signal. The labeling procedure is described in detail in Appendix B, building upon the notion and observations of (Schneebeli et al., 1999). The first labeling phase was conducted by one expert, and in the second phase, two experts revisited the profiles to ensure consistent labeling. The labeling process involves using Micro-CT samples and NIR photography to validate the snow types identified from the force signal where possible. When assigning the labels to the SMP profiles, we lean to the above-mentioned international classification guideline of seasonal snow on the ground Fierz et al. (2009). However, we regard the labels assigned to the SMP signals as mere *approximators*. During the labeling process, signal types are grouped together, and we infer from Micro-CTs which snow type matches each group best. Since we seek a language that is common to the snow community, we are using the labels provided by (Fierz et al., 2009) where possible. Since (Fierz et al., 2009) focuses on Alpine snow and does not cover all snow types on Arctic sea ice, such as different forms of "Depth Hoar", we extend those labels where necessary. The resulting labeled profiles were used during training, testing, and validation, while some unlabeled profiles were used for semi-supervised models and out-of-distribution tests. Up-scaling consistent labeling of SMP profiles is exactly the type of task that ML algorithms can tackle.

---

[1]Fierz et al. (2009) refer only to visually observed snow grains; not to SMP signals.

We preprocessed each SMP profile as well as the complete labeled dataset. The surface and the ground of the profiles were detected automatically by the snowmicropyn package [2]. For each SMP profile, we replaced negative force values with 0, summarized the signal into bins (1 mm), and added mean, variance, maximum, and minimum force values for those bins. Those values were also determined for a 4 mm and 12 mm moving window. Moreover, Löwe and Van Herwijnen (2012)' Poisson shot noise model was used to extract $\delta$, $f$, $L$ and the median force value for a 4 and 12 mm window. We added further depth-dependent information, including the distance from the ground and position within the snowpack for each data point. Refer to Table C1 in Appendix C for an overview of all features used for each SMP profile, and to Table C2 to see the feature importance for each snow type.

We preprocessed the complete labeled dataset by normalizing it, removing profiles from the melting season, and merging snow classes. For example, "Decomposed and Fragmented Precipitation Particles" are merged with the class "Precipitation Particles" since they represent a similar type of snow. The few occurring "Ice Formations" and "Surface Hoar" instances in the MOSAiC dataset are summarized in the class "Rare". While a high classification performance cannot be expected for the rare classes, we still include them to show how the models perform on a "real-world dataset" that in most cases will also include classes with few occurrences. The data preprocessing ensures that the dataset is clean and that all necessary information, such as depth-dependent information, is available during classification.

The resulting dataset has the following properties: (1) There are multiple, noisy, and overlapping classes. (2) There is a between-class imbalance, i.e. some snow types occur much more frequently than others. (3) There is a within-class imbalance, i.e. some grain classes contain different sub-grain-classes, but some of them are more frequent than others. (4) The labeling of classes is afflicted with uncertainty, i.e. snow experts themselves are not sure to which class exactly some data points belong. The complexity of the data set complicates classification and lowers the maximum achievable accuracy.

## 2.2 Task description

We compare the capabilities of different models to classify and segment the profiles of the MOSAiC winter SMP dataset. To this end, the models first classify each data point of the signal and then summarize the classified points into distinct snow layers ("first-classify-then-segment"). This task can be solved with different learning and classification techniques.

The task can be addressed via **independent classification** or **sequence labeling**. In independent classification, each individual point is classified independently, without looking at other data points. The underlying assumption is that each individual data point carries enough information to be classified solely on that basis. In contrast, sequence labeling assumes that the data is an intra-dependent sequence, where the label of each data point also depends on the preceding labels (Nguyen and Guo, 2007).

The models can follow either the **supervised**, **unsupervised**, or **semi-supervised learning** regime. In supervised learning, labels are provided to learn an input-output mapping function (Russell and Norvig, 2021). In unsupervised learning, patterns and structure are found in unlabeled data (Ghahramani, 2004), however, no classification is possible, which is why no unsupervised models are employed here. Instead, semi-supervised models are used, which are able to find structures in sparsely

---

[2]https://snowmicropyn.readthedocs.io/en/latest/

labeled data and leverage this information during classification. In the following, all models employed in this work are shortly presented and put in the context of their learning and task type.

## 2.3 Models

The **majority vote** classifier is used as the baseline for the performance comparison and simply predicts always the majority class ("Rounded Grains Wind Packed"). It satisfies the criteria that a baseline should not require much expertise, should be easy to build, and fast to evaluate (Li et al., 2020).

The **cluster-then-predict models** employed in this study, can be separated into three different semi-supervised and independent classification models. Unsupervised methods are used to find clusters in the dataset and subsequently, a supervised model is used to assign labels to the cluster (Soni and Mathai, 2015; Trivedi et al., 2015). As an unsupervised model, k-means clustering (Forgy, 1965; Lloyd, 1982), mixture model clustering (GMM) (Bishop, 2006) and Bayesian Gaussian mixture models (BGMM) (Bishop, 2006) were used. The supervised part of the model is a simple majority vote within the clusters, in order to see if the unsupervised model adds enough information to beat the majority vote baseline.

**Label propagation** is a graph-based, semi-supervised, independent classification algorithm. It propagates the labels of labeled data points to unlabeled ones (Zhu and Ghahramani, 2002). Here, a modified version of this algorithm by Zhou et al. (2003) is used (also known as "label spreading") (Yoshua et al., 2006; Pedregosa et al., 2011).

**Self-trained classifiers** turn a given supervised classifier into a semi-supervised independent classifier. It follows an iterative approach of training a supervised model on labeled data, predicting more data with the model, and retraining the model with the most confident predictions (Yarowsky, 1995).

**Random forests** (RFs) are ensembles of diversified decision trees (supervised and independent classification). The diversification happens via tree and feature bagging, where only subsets of data or features are used during training (Ho, 1995; Breiman, 2001). Decision trees are simple to build, explainable, white-box classifiers and for these reasons among the most popular machine learning algorithms (Wu et al., 2008). Additionally, a balanced random forest was used with random under-sampling to balance the data (Chao et al., 2004).

**Support vector machines** (SVMs) construct a hyperplane in a high-dimensional space to solve binary classification tasks (Cortes and Vapnik, 1995; Han et al., 2012) (supervised and independently). When a problem is non-linearly separable, the input data can be projected into a higher-dimensional space until the problem becomes linearly separable. The kernel trick can be used to circumvent the computationally expensive data transformation involved here. It directly extracts a non-linear optimal hyperplane (Schölkopf et al., 2002).

**K-nearest neighbours** (KNN) is a local, non-parametric classification method that compares samples and classifies new samples based on their $k$ nearest training data points (supervised and independently). The class of the prediction sample is determined via a majority vote. (Fix and Hodges, 1952; Cover and Hart, 1967)

**Easy ensemble classifiers** are ensembles of balanced adaptive boosting classifiers (supervised and independent). The method is especially helpful for imbalanced datasets since the learners are trained on different bootstrap samples, which are balanced via random under-sampling. (Liu et al., 2008)

**Long short-term memories** (LSTMs) are a form of artificial neural networks (ANNs) and can perform supervised sequence labeling tasks. ANNs incrementally update their decision function that describes the decision boundary between classes. ANNs have different nodes, which can be seen as representing different parts of the functions which are weighted differently. During training, the weights of the ANN are optimized by minimizing a loss function via gradient descent. A long short-term memory can handle time-series data. It consists of different memory cells so the LSTM can forget information that is no longer needed,

remember information that is required for future decisions, and retrieve information that is required for current decisions. (Hochreiter and Schmidhuber, 1997; Jurafsky and Martin, 2021)

    **Bidirectional LSTMs** (BLSTMs) connect two independent LSTMs where the first LSTM processes the inputs forward and the second one backwards. The outputs of both LSTMs are connected to one output. This architecture is helpful when the dependencies of a time series go in both time directions, which is the case for snow profiles. (Schuster and Paliwal, 1997;

Jurafsky and Martin, 2021)

    **Encoder-decoder networks** consist of an ANN encoder that compresses the time-dependent information into a vector and a decoder that uses this information to solve a supervised sequence labeling task. Additionally, the attention mechanism can be used to strengthen the ability to learn long-term dependencies by focusing only on the parts of the input sequence that are relevant for the current time step. (Bahdanau et al., 2014; Jurafsky and Martin, 2021)

## 2.4   Evaluation

In this work, (1) the performance of different models is compared, (2) differences in the classification of different snow types are analyzed, and (3) the generalization capability of the best-performing model is examined. (1) The performance comparison is done by looking at the metrics of each model and the specific predictions on the test data set. The metrics used here are accuracy, balanced accuracy, weighted precision, F1 score, area under the receiver operating characteristic (AUROC), log loss,

fitting, and scoring time (see Appendix D for further explanations). (2) The label-wise performance is analyzed with the help of label-wise accuracy plots and receiver operating characteristic (ROC) curves. ROC curves plot the true positive rate versus the false positive rate. The higher the area under the ROC curve, the clearer the model can separate between positive and negative samples. (3) The generalization capability is tested by running the best-performing model on 100 random profiles from different parts of MOSAiC winter data. These profiles are outside of the distribution of the training, validation, and testing data and we

refer to them as "out-of-distribution profiles". Here, the "out-of-distribution" profiles contain the same classes as the training data, so the model still has a chance to predict the correct labels. Evaluating these three aspects ensures that users can choose a model and know (1) how it performs compared to other models, (2) what to expect from the snow-type-specific predictions, and (3) how robust a chosen model will be.

## 2.5   Experimental setup

The experimental setup includes a training, validation, and testing framework: roughly 80% of the labeled dataset is used for training and validation, while the other 20% is set aside for testing. Validation is realized as 5-fold cross-validation (Stone, 1974). The hyperparameters were tuned on the validation data and the best-found hyperparameters were used during testing.

| Category | Model | Absolute Accuracy | Balanced Accuracy | Precision | F1 Score | ROC AUC | Log Loss | Fitting Time | Scoring Time |
|---|---|---|---|---|---|---|---|---|---|
| Baseline | Majority Vote | 0.39 | 0.14 | 0.15 | 0.22 | nan | nan | **< 1** | **< $10^{-3}$** |
| Semi-Supervised | K-means | 0.62 | 0.44 | 0.60 | 0.61 | nan | nan | 385 | 0.01 |
| | GMM | 0.65 | 0.36 | 0.57 | 0.61 | nan | nan | 151 | *0.008* |
| | BGMM | 0.65 | 0.38 | 0.63 | 0.63 | nan | nan | 225 | 0.009 |
| | Self-trainer | 0.69 | 0.67 | 0.74 | 0.71 | 0.92 | 0.84 | 19 | 0.29 |
| | Label propagation | 0.71 | 0.54 | 0.72 | 0.71 | 0.92 | 1.5 | 10 | 3.35 |
| Supervised | Random Forest | 0.73 | 0.60 | 0.73 | 0.73 | 0.93 | 0.70 | 72 | 0.97 |
| | Balanced RF | 0.70 | **0.67** | 0.74 | 0.71 | 0.92 | 0.84 | 9.9 | 0.58 |
| | SVM | 0.71 | 0.66 | 0.73 | 0.71 | 0.93 | 0.67 | 19 | 7.45 |
| | KNN | 0.71 | 0.54 | 0.71 | 0.71 | 0.89 | 3.58 | < 1 | 1.84 |
| | Easy Ensemble | 0.62 | 0.59 | 0.70 | 0.64 | 0.88 | 1.66 | 46 | 42.5 |
| ANNs | LSTM | *0.75* | 0.58 | *0.75* | *0.75* | **0.94** | **0.63** | 349 | 2.3 |
| | BLSTM | 0.74 | 0.58 | 0.74 | 0.73 | 0.93 | 0.79 | 975 | 3.4 |
| | Encoder-Decoder | **0.78** | 0.54 | **0.78** | **0.77** | *0.94* | *0.64* | 2911 | 5.8 |

**Table 1.** Results of different models from the categories baseline, semi-supervised, supervised and ANNs. The best values among all models are **bold**. Second-best values among all models are *italic*. The best values among one category are underlined. ROC AUC and logistic loss (log loss) could not be determined for the baseline and some of the semi-supervised models due to the design of these models.

Hyperparameter tuning is the process of searching the optimal internal learning settings of an ML model. Hyperparameters control the learning process of the models, whereas parameters are learnt by the model. The tuning is performed on the validation data and the hyperparameters that achieve the highest performance for their model chosen for subsequent model evaluation. Here, tuning was applied moderately and with a simple grid search. All tuning results can be found in the GitHub repository. Specifications of the machine on which the experiments were run can be found in Appendix E and descriptions of the model setup can be found in Appendix F.

## 3 Results

### 3.1 Classification performance of models

Overall, the results show that an automatic classification and segmentation of SMP profiles with ML algorithms is possible, even if no further information such as snow-pit data or manual segmentation is provided. Category-wise all semi-supervised

models were not performing particularly well (see Table 1). Only the self-trainer could compete with models from other categories, but this might be the case because the self-trainer is based on a balanced random forest. The supervised models achieved mixed performances: Some models such as the random forests and the SVM are clearly performing well, whereas other models such as the KNN and the easy ensemble are underperforming. Overall, the random forest was the best model in the supervised category since it achieves the highest absolute accuracy (0.73) and F1-Score (0.73). However, considering rare classes, the balanced random forest outperformed the plain random forest. All three ANNs did exceptionally well and their category was clearly the most successful among all three categories. The encoder-decoder showed the best scores among all models in terms of absolute accuracy, precision, and F1-Score, closely followed by the LSTM. We consider the LSTM the best model within that category since the encoder-decoder only reached its high performance after extensive hyperparameter tuning and underperformed significantly when not tuned well. In contrast, the LSTM achieved its performance more consistently and even under moderate hyper-parameter tuning, and is thus more suitable for users. The subsequent analyses compare those three models that performed best within their category: the LSTM performed best among the ANNs, the random forest among the supervised models, and the self-trainer among the semi-supervised models.

Different ML models exhibited different prediction styles in terms of smoothness and ability to predict rare classes. In Fig. 2 it becomes visible that the models' predictions are not far off from the labels. In general, the predictions are somewhat similar to the labeled profiles but the models often had difficulties in determining the precise start and end of a segment. Looking at three random exemplary profiles of the test data in Fig. 3, one can see that the three main models seem not only to generate similar predictions but make also similar mistakes. In the medium-deep profile (middle column), all three models predicted a longer segment of "Depth Hoar" that was actually not present in the labeled profile. In the shallow profile, all three models predicted some intermediate "Depth Hoar Wind Packed" layers in the first third that did not exist. And in the deep profile, all three models miss the narrow intermediate "Depth Hoar" layer. In summary, it becomes apparent that the different models are producing consistent predictions to a certain degree. Of course, there are significant differences among the models, too. First of all, the LSTM is closest to the labeled profiles (see Fig. 3). Secondly, the LSTM provided much smoother and less fragmented predictions than the other two models. And thirdly, the self-trainer clearly overestimates rare classes, which hurts the overall performance. To summarize, the LSTM, random forest, and self-trainer show certain prediction similarities among each other, however, the LSTM imitates expert labeling best.

## 3.2 Classification difficulty of snow types

Fig. 4 shows that some snow types are easier and others are harder to classify. The label-wise accuracy seems to be influenced by the following factors: (1) choice of model, (2) frequency of snow type in the dataset, (3) snow type itself. Within one snow type category, the models perform differently well, however, some snow types seem to be easier, and others are more difficult to classify for all models. For example, "Rounded Grains Wind Packed" achieved a high accuracy among all models, whereas "Depth Hoar Wind Packed" achieved a low accuracy among all models. This could be partially attributed to the fact that there are fewer samples available for "Depth Hoar Wind Packed". However, the snow types themselves seem to influence the classification difficulty as well: the class "Precipitation Particles" achieves high accuracy values among some models, despite

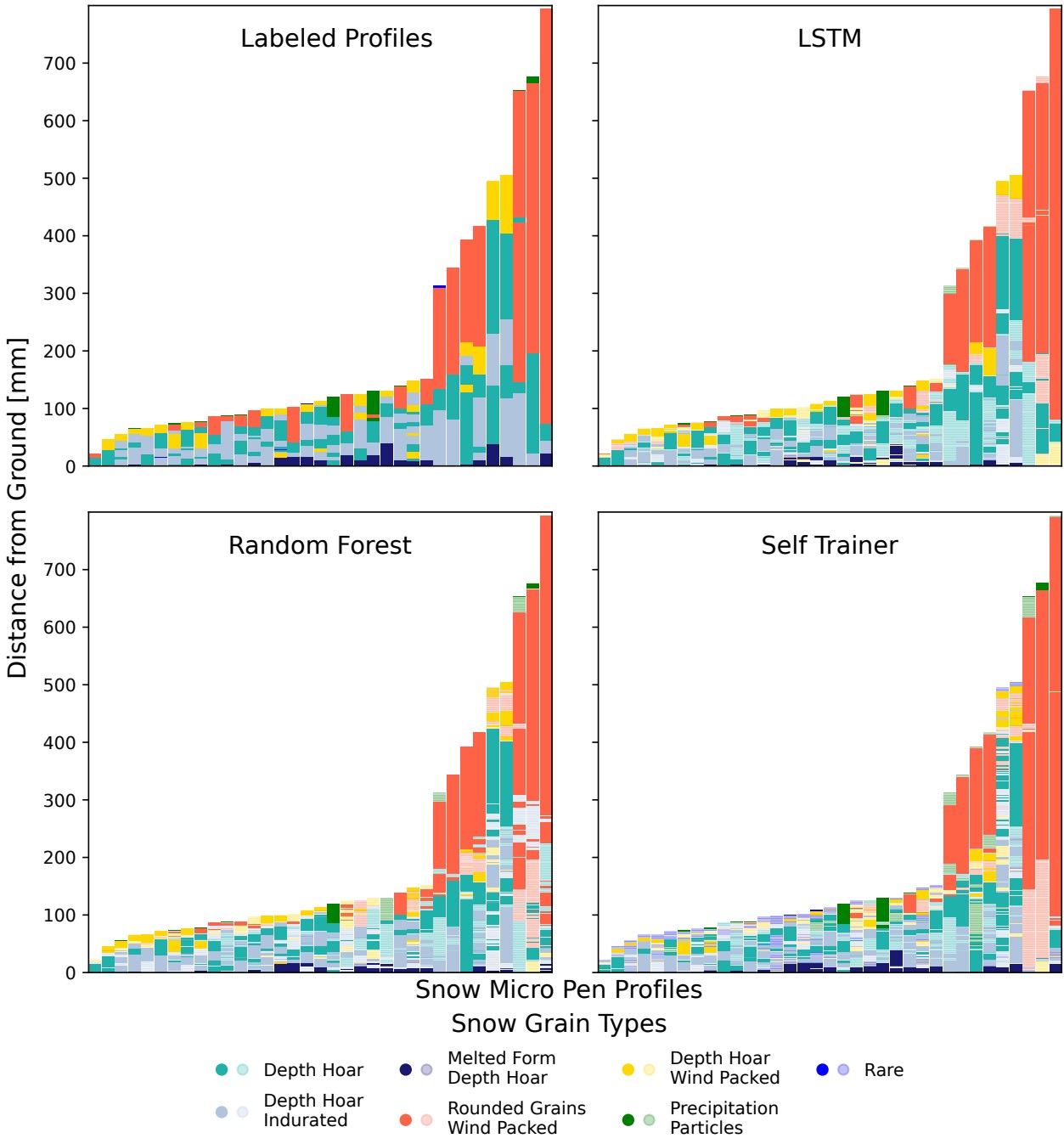

**Figure 2.** Predictions on the test dataset of the LSTM, random forest, and self-trainer. The upper left panel shows the labeled data. In the other panels, the correct predictions are shown with more intense colours and the wrong predictions with less intense colours. The LSTM has the highest rate of correct predictions and imitates the smoothness of the labeled data very well. The random forest does well but provides more segmented predictions. The self-trainer immensely overestimates rare classes.

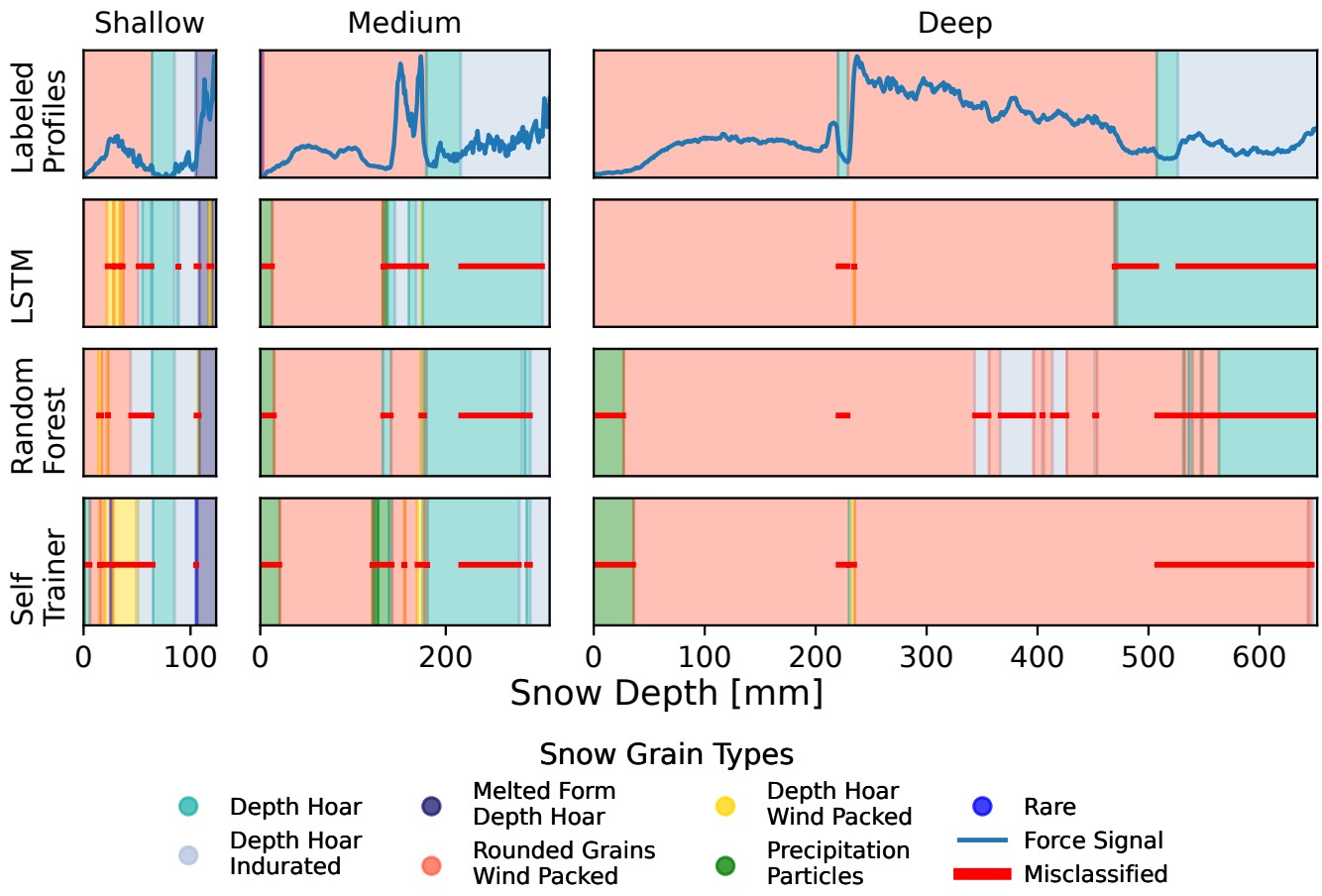

**Figure 3.** Model predictions for three randomly chosen SMP profiles. The first row represents the labeled profiles (with force signal). The subsequent rows represent the LSTM's, random forest's, and self-trainer's predictions, with the red bar indicating wrong predictions. Each column shows a different profile randomly chosen from the test data (shallow profile: S31H0276; medium profile: S31H0206; deep profile: S49M1918). All three models seem to make similar mistakes, e.g. they predict a larger portion of "Depth Hoar" at the end of the medium SMP profile. The predictions of the LSTM are closest to the labeled profiles.

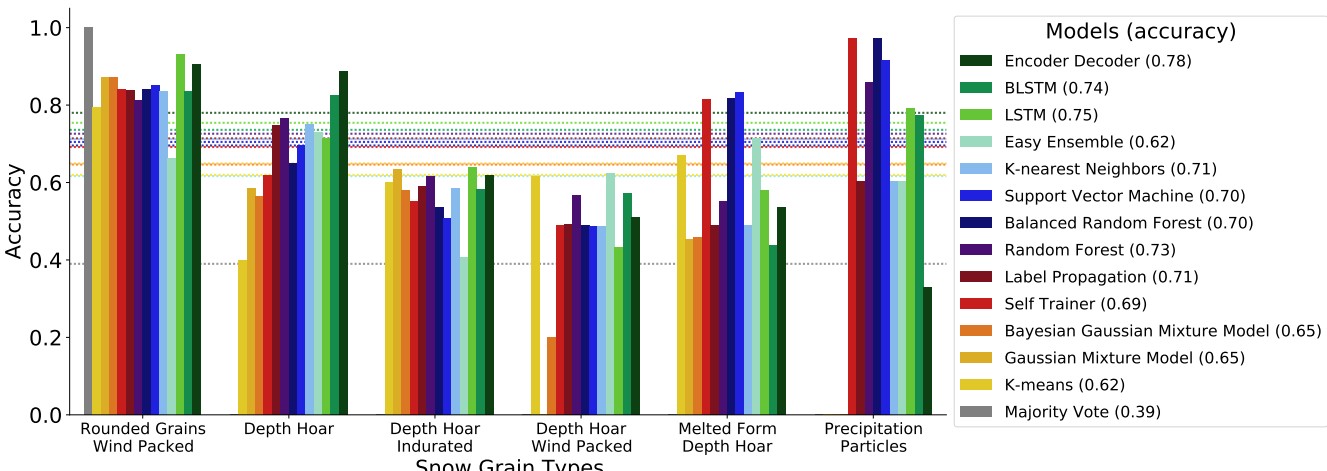

**Figure 4.** Label-wise accuracy of all models. each model is encoded with a different colour. The most frequent label is on the left of the x-axis ("Rounded Grains Wind Packed"), and the least frequent is on the right ("Precipitation Particles"). The class "Rare" was dropped. Each bar represents the accuracy for a single snow type. The dotted lines show the overall accuracy performance of each model. The encoder-decoder, the BLSTM, and the LSTM achieved the highest accuracy values. For all models, some classes are more difficult to classify than others: e.g. "Depth Hoar Indurated" and "Depth Hoar Wind Packed". Some classes are easier to classify than others, such as "Rounded Grains Wind Packed". Some classes can only be classified well by a subset of the models, such as "Precipitation Particles" and "Melted Form Depth Hoar".

the fact that it is the rarest class in the dataset. For some snow types, some models are able to access certain information enabling a high performance on that particular snow type – independent of its frequency. This means that the classification difficulty does not only depend on the number of available samples. Instead, several other underlying characteristics determine the classification of difficulty of each snow type as well, most notably: (1) The initial classification, which is not always completely consistent; (2) the underlying micro-mechanical properties, i.e. some snow types have characteristic force signals that separate them more clearly from others; (3) the training data set since it does not cover all types of force signals.

Depending on the model, a higher accuracy score could lead to a lower precision score for a label (accuracy-precision trade-off). The ROC curve in Fig.5 illustrates this relationship between the true positive and false positive rates for the different snow types and their averaged performances. It becomes apparent that both the snow type and the choice of model influence the accuracy-precision trade-off. The class "Rare" for example seems to be difficult to classify both accurately and precisely for all models, whereas "Precipitation Particles" are showing an almost perfect ROC curve. If one is interested in choosing a model that performs well for a particular snow type, these ROC curves can reveal which model is most suitable. To get even more detailed label- and model-wise insights, refer to the confusion matrices in Appendix H. Both the LSTM and the random forest achieve an area under the ROC curve of 0.96. However, on average (see Fig. 5, pink dotted line), the LSTM outperforms the self-trainer and random forest and is thus most suitable for general classification tasks.

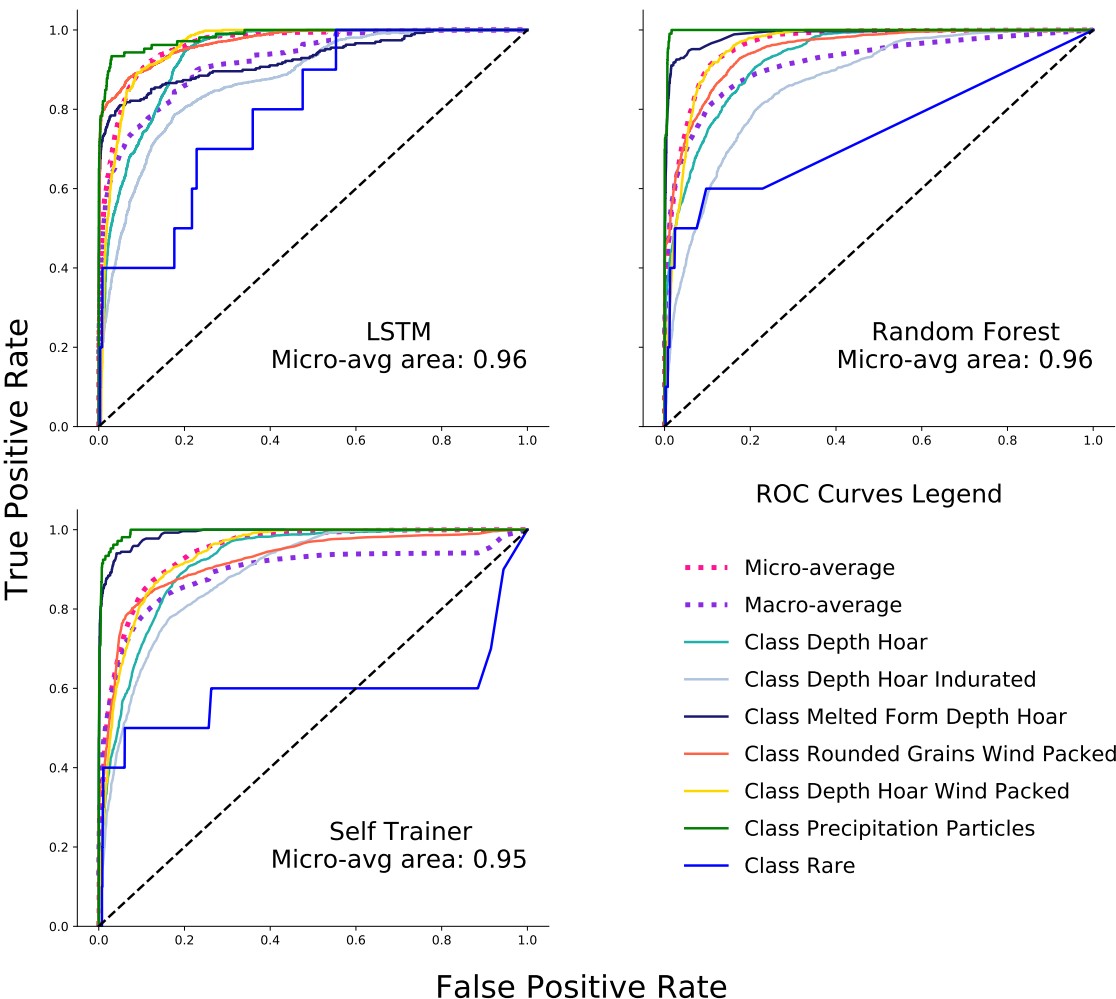

**Figure 5.** ROC curves of the LSTM, random forest, and self-trainer for each class. The dotted lines are the micro- and macro-averaged ROC curves. The macro-average calculates the ROC for each class and averages the performances afterwards. The micro-average weights the performance according to class contribution (balanced performance results). The LSTM achieves the highest ROC performance overall. The order of the best-performing snow types is similar among all models. The classes "Rare" and "Depth Hoar Indurated" have the lowest ROC areas, whereas "Precipitation Particles" has the highest ROC area for all models.

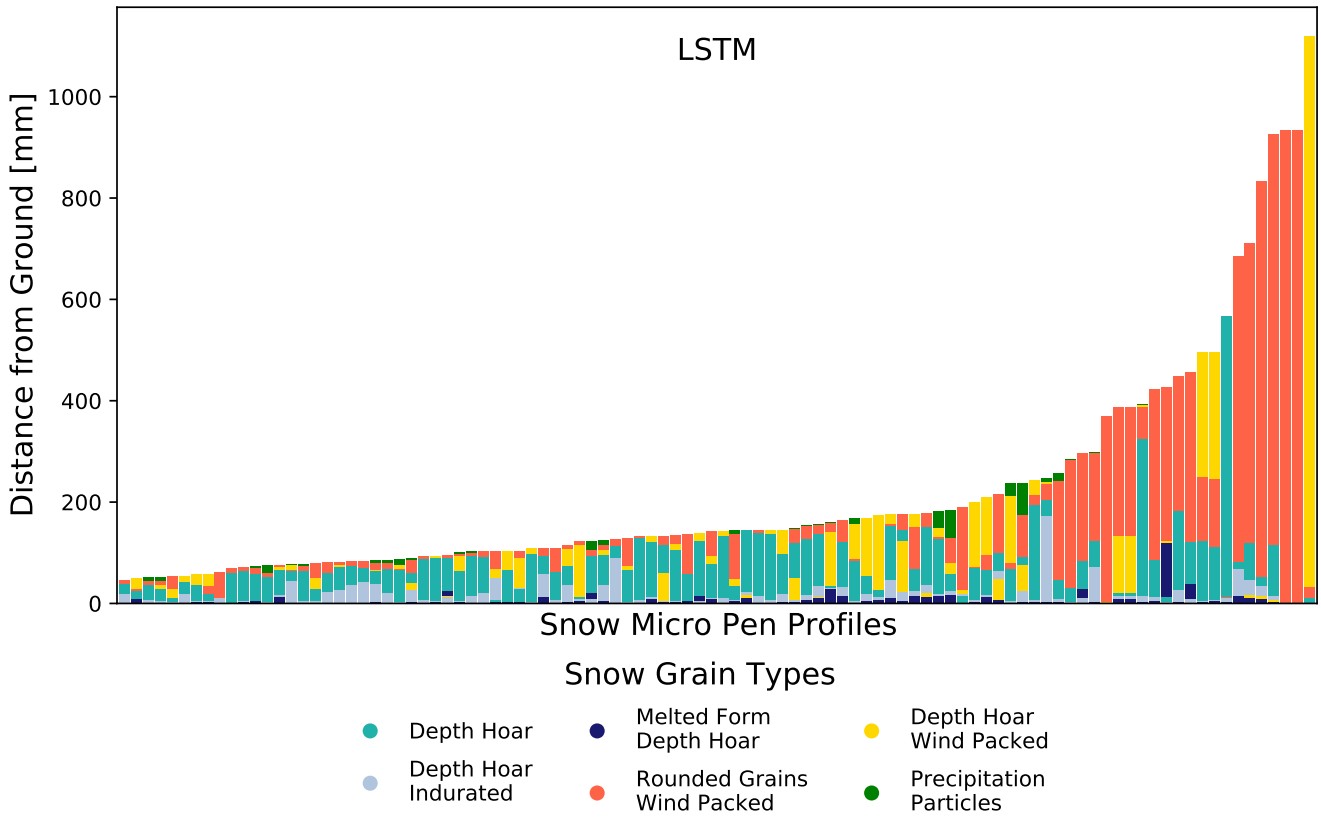

**Figure 6.** LSTM SMP profile predictions on out-of-distribution data. The SMP profiles used here come from different legs of the MOSAiC expedition than the training, validation, and test data. The profiles used here still stem from the winter season to ensure that the same set of snow types can be used as in the training dataset. The distribution of the predicted profiles looks convincing, with only a few profiles standing out as certainly wrong predictions (e.g. most right profile with $\sim 90\%$ "Depth Hoar Wind Packed").

## 3.3 Generalizability

The prediction of the LSTM for 100 random profiles outside of the training and testing distribution is shown in Fig. 6. Since the labeled profiles are not yet available for these predictions, the generalization capabilities can only be evaluated on the basis of what seems "reasonable". "Melted Form Depth Hoar" appears only at the ground of the profiles, "Precipitation Particles" only at the top, "Rounded Grains Wind Packed" are mostly at the top and rather deep – these are all "reasonable" predictions. However, there are also some predictions that are not reasonable or at least unexpected: the left profile consists almost entirely of "Depth Hoar Wind Packed", sometimes "Depth Hoar Wind Packed" appears right before "Melted Form of Depth Hoar", and "Rounded Grains Wind Packed" sometimes appear briefly in the "middle" of a profile (and not at the top). Overall, the LSTM seems to make mostly reasonable predictions, however, an in-depth expert analysis of the predictions is necessary to validate that further.

## 4  Discussion

The results showed that the automatic classification of SMP profiles is possible with up to 78% accuracy. In the following the nature, impact, and limits of these results are discussed.

The metrical results presented are in line with previous findings: King et al. (2020a) reported an overall accuracy score of 0.76 when using SVMs and additional snow pit information to classify three snow types. Satyawali et al. (2009) achieved an average accuracy of 0.81 when using the nearest neighbour approach and knowledge rules to classify five snow types. However, these results stem from only three profiles and are not representative. Havens et al. (2013) achieved an accuracy of maximal 0.76 (global dataset) when using random forests and time-intensive manual layer segmentation to classify three snow types. The major difference from these previous results is that the accuracy results of this study were achieved for *seven* snow types,

without time-intensive layer picking, snow pit digging, or additional knowledge rules. This means that in contrast to previous work, the models here can be directly employed by users for their own SMP datasets in the field: simply retrain and predict. For this, they only need to provide a set of training samples for their specific dataset and classification style. The work presented here enables scientists for the first time to rely on fully automated ML SMP profile classification and segmentation.

The results were also satisfying to domain experts since the predictions were in themselves consistent and followed the patterns of the training data. In general, the snowpack on sea ice is extremely variable, and the traditional snow types are often a mixture of different features. This becomes visible when comparing the SMP-profiles to the micro-CT samples. In the view of the authors, a temporally consistent classification is more relevant to the interpretation of the development of the snowpack, even if there is a certain, but unknown, bias to an expert interpretation. Hence, the models were also in practice helpful to

analyse Arctic snowpack development.

### 4.1  Classification performance of models

Each model category performs differently because each model takes different aspects of the data into account. Semi-supervised models try to take unlabeled data into account to improve their predictions, however, this did not work well in our context. The most likely reason for the overall underperformance of this category is that the unlabeled data contained out-of-distribution

data, i.e. the unlabeled data had different underlying mechanisms than the labeled data (different parts of the winter season). Another reason might be that only a small subset of unlabeled data was included in order to limit running times. Moreover, the poor performance of the cluster-then-predict models is most likely also a result of the classifier used after clustering: a more sophisticated method than a majority vote classifier is needed here.

The simple supervised models take one data point after the other into account and do not consider time-series structures

within the data. The algorithms used in all previous SMP automation studies fall into this category. In contrast, ANNs are supervised models that take the underlying time sequence of the data into account. While the supervised model in general performed well, they were still clearly outperformed by the ANNs. A likely reason why the ANNs outperformed all the other models is precisely the ANNs' ability to process time-dependent – or in the case of snow profiles depth-dependent – information. ANNs are tackling the classification task as a sequence labeling task which enables them to include information from the

order and position of snow layers. The supervised models still have access to time-relevant information (time-window features, see Appendix C1), however, they do not have any ability to learn time-based information (what should be remembered and forgotten). Besides, the ANNs learn to imitate the training set, leading to smooth and expert-simile predictions. In comparison, taking the time component of SMP signals into account has not been done in previous methods and we argue that it adds a major information piece and boosts the overall prediction performance significantly.

Each model exhibits a different prediction style due to the models' intrinsic differences and thus might be suitable for specific tasks. The following aspects are listed for consideration (user's guide):

A **Time and resources for hyperparameter tuning.** The LSTM and the encoder-decoder network are recommended when plenty of tuning time is available. Especially, the encoder-decoder network performs badly if not tuned well. The SVM and the balanced random forest need little tuning time, whereas the random forest is the go-to model in case (almost) no tuning time can be provided.

B **Need for a simple to handle, off-the-shelf algorithm.** Among the high-performing models, the random forest and the SVM are the easiest to handle off-the-shelf algorithms. The self-supervised algorithms and especially the ANNs require a somewhat deeper understanding of the models and the ability to implement them.

C **Desired level of explainability.** The random forests are most explainable since the decision trees can be directly visualized (Appendix G). The ANNs are the least explainable models (without further modifications).

D **Importance of minority classes.** When deciding on a model, the underlying task must be examined as well: In the case of avalanche prediction, it might be essential to predict a buried layer of "Surface Hoar", a very rare class, which needs to be detected no matter the costs. In such a case of "minority class prediction," the balanced RF or the SVM should be employed. The ANNs and the random forest, in contrast, are more suitable to achieve an overall good classification.

E **Availability of unlabeled data that is from the same distribution as the labeled data.** In case a lot of unlabeled data from the same distribution and time is available, the self-trained classifier can be considered. The weak learner of the self-trained classifier can be chosen according to the criteria listed above. Since in this work we only had a small subset of unlabeled data stemming from the same distribution as the labeled data, further evaluations on the self-trained classifier and label propagation remain open.

This highlights that there is not a single best model, but instead, users can deliberately choose a model that suits their needs, such as overall accuracy, ability to predict rare classes, explainability, training, and deployment time.

## 4.2 Classification difficulty of snow types

Snow types are differently difficult to classify since their categories are rather continuous than discrete. This was also observed in previous work and in all previous works performances were reported label-wise to account for those differences (Satyawali et al., 2009; Havens et al., 2013; King et al., 2020a). We performed t-distributed stochastic neighbour embedding (t-SNE)

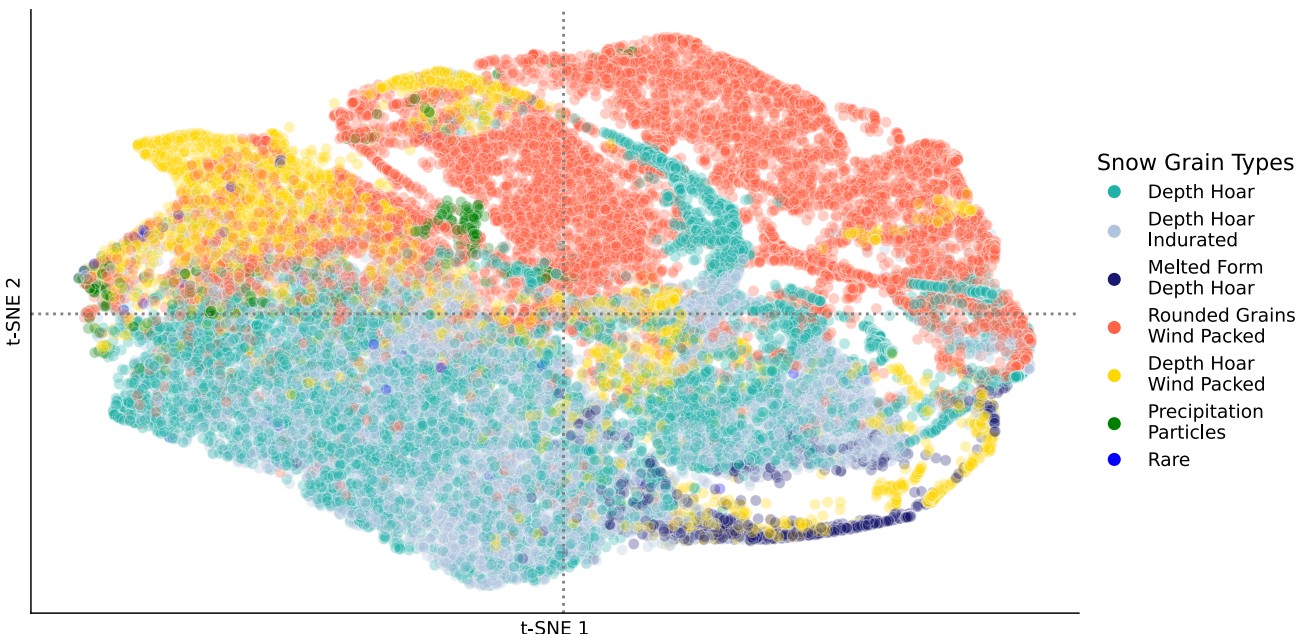

**Figure 7.** 2-dimensional t-distributed stochastic neighbour embedding (t-SNE) of SnowMicroPen (SMP) dataset. The colours encode the snow types. The figure shows that (1) "Depth Hoar" and "Depth Hoar Indurated" are hardly separable, (2) "Depth Hoar Wind Packed" is similar to several other snow types, and (3) "Precipitation Particles", "Melted Form of Depth Hoar" and "Rounded Grains Wind Packed" can each be separated more clearly from the other snow types.

on the SMP dataset to visualize how separable the different classes are (see Fig. 7). "Precipitation Particles", for example, appears as a singled-out green grouping, which is in line with our and other findings (Satyawali et al., 2009) that it is easier to classify than other snow types. We conclude that some classes have features distinguishing them more strongly from other snow types. The class "Rounded Grain Wind Packed" behaves similarly (Satyawali et al., 2009). However, some classes, such as "Depth Hoar" and "Depth Hoar Indurated" are completely overlapping in Fig. 7, and indeed our models had problems with differentiating between those two classes. Similarly, "Depth Hoar Wind Packed" seems to overlap largely with "Rounded Grains Wind Packed" and "Melted Form of Depth Hoar". We theorize that the reason for their non-separability is that those snow types transform into each other during snow metamorphosis. This means many data points can not be discretized into one single category since they are on a continuous spectrum. Satyawali et al. (2009) pointed out, as well, that they often found data points being in transition between snow classes and attributed it to the fact that the snow is changing continuously. In conclusion, it is currently impossible to reach 100% classification accuracy on every snow type since some snow types will always lie between categories.

Despite these difficulties, the underlying SMP signals are still characteristic enough for specific snow types to be classified successfully. The different micro-mechanical properties of the snow types are reflected in the SMP signal and are thus the

385 driver for the classification. Some classes, such as "Precipitation Particles", can be clearly separated from others since the bonding between the grains is so weak that the force signal is very low. As long as "Precipitation Particles" are not sharing this characteristic with other snow types, they can be easily classified. Refer to Appendix B to learn more about the relation between snow types and SMP signal, and refer to Appendix G to see which classes have unique and which classes have shared signal characteristics.

The classification difficulties also extend to the expert labeling process itself. The continuous nature of the snow types makes it particularly difficult for domain experts to agree on labeling, i.e. two different snow experts will produce two different labeled and segmented profiles for the same SMP measurement (Herla et al., 2021). This is another reason why a classification accuracy of 100% cannot be reached. One might suggest supplementing the classification process with additional observational data to make the process more "objective", as we also do here. However, each classification and segmentation of a snowpack is
"subjective" in nature right now, no matter which observational data is used as the basis for the classification. When requesting a segmentation and classification of a snowpack, one is always requesting the classification of a specific expert. While the operator bias can be mitigated by using NIR, Micro-CTs, or the SMP, the classification of those measurements remains subjective. It is neither this study's goal nor task to provide an objective classification; instead, we aim for a *consistent* classification.

Difficulties in reaching 100% accuracy do not preclude overall good performance, however. While experts may end up with
400 different segmentations and classifications, they can still agree that two different analyses are both valid analyses of the same profile. Similarly, the algorithms provided here output predictions that may not always align with the expert labeling but are sensible and directly usable. Hence, we cannot evaluate the models solely based on numerical metrics such as accuracy but must also evaluate the performance from a qualitative perspective. This is the reason why we evaluated if an SMP user, who also labeled the training data, would (1) accept the predictions of the ML algorithms on an out-of-distribution dataset, (2) find
them consistent with their own labeling, (3) and would subsequently work with those predictions. In the case of the MOSAiC dataset, all those aspects were fulfilled. We find such a qualitative assessment important since these questions decide whether or not the tools provided will be used in practice.

We further want to point out that the algorithms themselves are entirely agnostic to the question of "subjectivity". The algorithms are merely reproducing what they have been trained on. If we can provide the algorithms with a dataset that can
be considered "fully objective" and the community agrees on that as ground truth data, the algorithms could reproduce those hypothetical "objective" labels. Alternatively, signals could also be grouped first, and some abstract classes could be assigned to them. Nevertheless, even this would rely on human expertise since the parameters to separate those groups would be subject to discussion (see Figure 7: The groups are not simply separable from each other, and the clustering would depend on parameter choices). In general, we provide a methodological framework here to classify and segment SMP profiles –which classification
patterns are reproduced depends on the user's choice.

The benefits of using an automatic classification are that the SMP user can (1) save valuable time, (2) receive consistent labeling, and (3) perform statistical analysis on their SMP dataset. In the case of the MOSAiC dataset, manual labeling would have meant labeling over 3000 profiles, which can easily take up to a year to classify (next to other obligations of domain experts). In terms of consistency, we already experienced how some of the models' predictions helped us –to our surprise

–to detect human mistakes and inconsistencies during the first labeling round. Furthermore, such an up-scaled classification enables, for the first time, the statistical analysis of an SMP dataset. One of the initial research questions for MOSAiC was "Is Depth Hoar in Arctic snowpacks mostly present at the bottom and Rounded Grains Wind Packed at the top?". With the help of snowdragon, the MOSAiC dataset could be enough consistently and accurately labeled to answer such a question with "Yes, this is indeed the case.".

## 4.3   Generalizability

The LSTM can generalize to other winter profiles with the same snow types since the underlying classification and segmentation rules stay the same. However, the LSTM's generalization capability does not extend to other seasons or regions when / where other snow types are found, such as melted forms or regional snow types. As mentioned before, the models do not generalize on different classification styles of experts. The models used in this work are still generalizable in that they can be used on any desired dataset as long as they are re-trained on the chosen dataset. This would not have been possible in previous works such as Satyawali et al. (2009) since knowledge rules for one snow region and season do not transfer to other regions or seasons. For greater generalization capability, the LSTM – or any other model – must be either trained with a more general dataset or must be specifically re-trained for an individual data set.

## 4.4   Limitations and future work

As previously discussed, the uncertainty of expert labeling is a general limitation of this particular study. While this uncertainty might be partially mitigated further by using a dataset for which many additional in-situ observations exist, it would still remain an issue. One approach for future work would be to quantify the uncertainty that is inflicted upon the labeled profiles. Subsequently, a machine learning model could be trained to classify not only snow types but provide a *probabilistic* classification.

This work does not address the task setting of first-segment-then-classify because this would require a completely different set of methods. In a first-segment-then-classify setting, the SMP signal could first be segmented with techniques used in audio-segmentation (Theodorou et al., 2014). The resulting time-series pieces could subsequently be classified as a whole (Ismail Fawaz et al., 2019). Future work could experiment with this problem formulation and analyze if performance further increases in this setting.

The ANNs used here are off-the-shelves and are not adapted to the specific underlying task in order to ensure a fair comparison between the different models. However, one could look into adapting the loss functions to include similarity measurements between snow samples. Results from clustering, performed on t-SNE data, could then be leveraged during classification to increase classification performance. Adapting the loss function of the ANNs could increase prediction performance greatly, however, such a loss function must be carefully constructed and evaluated on different datasets.

As mentioned in Sect. 4.3, the models cannot generalize to completely different settings in terms of seasons and regions. To ensure generalization capability one could train a large model on a dataset that includes snow types from different regions and seasons. Such a data set would need to be newly compiled because common SMP datasets are usually limited to one region

(Ménard et al., 2019; Calonne et al., 2020). In theory, a large enough model trained on a large enough dataset could be able to produce direct predictions for any SMP users. Thus, it would be interesting to train an ML model on a generalized dataset and validate its' performance on the specialized MOSAiC SMP dataset. This would shed new light on the spatiotemporal transferability of the ML models presented here.

Alternatively, SMP users can simply re-train a chosen model for their particular dataset. They need to provide a set of SMP profiles for their region, season, and classification style, but the overall time savings are still immense. To summarize, the generalization capabilities may be enhanced by using a more general dataset or one bypasses this problem by re-training to specific datasets – the snowdragon repository addresses the needs of the latter.

An immediate consequence of this study is the further analysis of the unlabeled part of the MOSAiC dataset. Domain experts can use the LSTM, or other models, to create predictions for the remaining 3516 profiles. A previously almost impossible task to classify and segment those thousands of profiles became feasible by providing just a set of 164 labeled profiles. The results of these predictions and their impacts on the cryospheric analysis of snow coverage in the Arctic will become apparent in future publications.

## 5 Conclusions

Snowdragon provides SMP users with a way to up-scale manual SMP labeling and provide large statistically consistent datasets. We showed for the first time that SMP profiles straight from the field can be automatically segmented and classified (up to 0.78 accuracy). Fourteen different models were trained here to classify seven snow types without providing any additional manual information. It also showed for the first time how ANNs and semi-supervised models can be used for the task of SMP classification and segmentation. Among all models, the LSTM and the encoder-decoder are performing the best. The resulting predicted profiles show smooth segmentations and expert-simile classification patterns that were satisfying to domain experts.

These findings will enable SMP users to automatically analyze their SMP measurements. To that end, an SMP user must simply decide on one of the fourteen models provided by the snowdragon repository, given the considerations listed in this paper, and retrain the model for their particular dataset. Afterwards, the SMP user can simply predict SMP classifications for the remaining unlabeled profiles.

The models presented here, in particular the LSTM, could be trained on a broad dataset from different regions and seasons so that automatic SMP classification becomes even more accessible. Such a model could even be integrated into the snowmicropyn package. The resulting tool would make knowledge about snowpacks easier and faster access for all scientists. This is of particular interest (1) for interdisciplinary scientists who rely on snow type information but do not have the tools to classify them themselves (remote sensing), (2) for scientists that require fast analysis of SMP profiles, such as in avalanche prediction and (3) for SMP users facing large datasets.

Snowdragon enables the analysis of the SMP MOSAiC dataset, a dataset containing detailed information about snow on Arctic sea ice. In times of climate change, this information is crucial: We need to understand the state of the sea ice in order to understand which state the Arctic system is in. For the first time, MOSAiC enables the scientific community to have access

to such a detailed and large dataset. And snowdragon is one example of how ML can help us to actually access the *knowledge* behind all the data.

*Code and data availability.* The current version of snowdragon is available on GitHub: https://github.com/liellnima/snowdragon under the MIT licence. To run the code version used in this paper, please refer v1.0.0 on GitHub or Zenodo: https://doi.org/10.5281/zenodo.7335813.
The exact version of the models used to produce the results used in this paper is also archived on Zenodo: https://doi.org/10.5281/zenodo.7063520 (Kaltenborn et al., 2022). The MOSAiC SMP data used as input and training data is available on PANGAEA: https://doi.pangaea.de/10.1594/PANGAEA.935554 (Macfarlane et al., 2021).

## Appendix A: User's Guide

Here, we provide a walk-through on how to use snowdragon with SMP profiles collected in the field.

1. Data collection

   – Collect the desired SMP profiles.

   – If you are familiar with snow stratigraphy measurements: Consider collecting additional in-situ observations such as Micro-CTs, NIR photography or similar to inform your labeling procedure. (see also points listed under "labeling").

   – If you are not familiar with snow stratigraphy measurements: Ask experts if a labeled dataset for your snow conditions exists (e.g. Macfarlane et al. (2021); Wever et al. (2022); King et al. (2020b) are publicly available) or if you need to onboard an expert to conduct a few in-situ observations and label some of your profiles.

   2. Labeling

   – Evaluate the following questions *before* you start the data collection.

   – If you conduct your own labeling:

– Use additional in-situ observations to fine-tune your labeling where possible.

      – Ask a fellow researcher for their opinion on a few profiles (before you label all of them).

      – Note down your labeling criteria - this way you can ensure consistency in your labeling.

      – Revisit your labeled profiles (all of them!) at least a second time. This way you can catch mistakes and ensure once more consistency in your labeling.

– If a labeled dataset exists for a specific location: Analyse carefully if the labeled data does transfer to your snow conditions. Can you expect the same snow types? Was the data collected in the same/similar location? Is it the same season? Might changing climatic conditions have also changed the nature of the snowpacks? Has the environment of the location gone through other types of changes?

- If labeled datasets exist capturing SMP profiles in general: Analyse carefully if you can work with a general dataset or need a specialized labeled dataset. Does the general dataset reflect the profiles you have collected well? Do you have snow types dominating your dataset that are a minority in the general dataset? Do you have a particular season dominating your dataset that is underrepresented in the general dataset? Does the general dataset contain all snow types that you have encountered in your dataset?

3. Set-Up

- Raw-Preprocess your SMP profiles and labels if necessary; data must be provided in `.pnt` format.

- Establish a consistent naming convention for your profiles. The labeling files (in `.ini` format) should have the same file name as the SMP profile that belongs to that labeling file. For example, you can have a `S31H0370.ini` containing the label markers for the force file `S31H0370.pnt`.

- Clone or fork the snowdragon repository: https://github.com/liellnima/snowdragon.

- Follow the setup guide in the GitHub repository.

- Tell the repository where your raw data lives: Change the `SMP_LOC` in `data_handling/data_parameters.py` to the right path as described online.

- Preprocess all the SMP profiles (follow online guidelines).

4. Model Selection

- Select the right model for your use case. Refer to Section 4.1 for further information.

5. Training and Evaluation

- Refer to the online guide of the repository.

6. Tuning

- Refer to the online guide of the repository.

7. Inference

- Use the `predict_profile()` or `predict_all()` functions from the `predict.py` file (provide path to data again). The functions can either be directly used or further adapted to your particular needs. The model you choose for inference must be stored somewhere, meaning you either need to train it beforehand or download the pre-trained models we provide.

8. Analysis

- Conduct your specific analysis on the labeled profiles. Run visualizations if desired as explained in the online guide.

## Appendix B:  Labeling

A snow micro penetrometer (SMP) is a device used to determine bond strength between internal snow grains in a snowpack. The micro-structural and micro-mechanical properties of the snow, for example, density and specific surface area (SSA), are directly influencing the bond strength. When a snow-micro penetrometer penetrates the snowpack and breaks these bonds between the snow grains, we are able to directly infer these micro-structural properties, as shown in the existing method by (Proksch et al., 2015). For example, snow with high density has a higher bond strength and therefore a higher penetration resistance force (measurable with the SMP), in comparison to low-density snow.

Different types of snow (Fierz et al., 2009) are known to have different densities and SSA, so the extraction of this data from the SMP force signal already allows us to draw pivotal conclusions about the snow type. However, the characteristics (using magnitude, frequency, and gradient) and the signature of the penetration force signal can provide more information about the internal snow type. This document outlines the process of classification of a snow type found on sea ice in the high Arctic using the SMP penetration resistance force signal.

Typical grains observed as part of the MOSAiC expedition on sea ice in the high Arctic are listed below.

- Precipitation particles (PP)/ Decomposing and Fragmented precipitation particles (DF)

- Ice formations (IF)

- Surface hoar (SH)

- Rounded grains, wind packed (RGwp)

- Depth hoar (DH)

- Depth hoar, indurated (DHid)

- Depth hoar wind packed (DHwp)

- Melt form, depth hoar (MFdh)

It is important to mention that the melt season is not included in this study due to liquid water influencing the interpretation of the SMP signal. For more information on the environmental and meteorological conditions under which the dataset has been collected refer to Rinke et al. (2021).

For the majority of snow types, we follow the classification of Fierz et al. (2009). However, Fierz et al. (2009) was adapted for Alpine snow, meaning some of the snow types listed above are either not included in the classification or differ from the ones encountered in Alpine snow.

**Melt form, depth hoar.** When working on sea ice we identified one alternative snow grain class (Melt form/ depth hoar, MFdh) that is not existing in the Fierz et al. (2009) classification. This snow type is known in the sea ice community as a surface scattering layer (Light et al., 2015). It is typically found in the summer season when sea ice melts, however, we identified this

as a persistent layer when transitioning into winter. In the field, this was an extremely dense layer at the snow-sea ice interface, and the penetration resistance force of this layer varied throughout the season. The label "melt form depth hoar" was chosen as this is a feature of melting sea ice that has persisted into the winter and has undergone metamorphism when buried under snow.

**Depth hoar, wind packed.** Initially wind packed rounded grains (RGwp) metamorphoses into a very hard, dense depth hoar under the large temperature gradients, which we call wind packed depth hoar (DHwp) (Pfeffer and Mrugala, 2002).

All other classifications are listed in (Fierz et al., 2009).

## B1 Classification details

| Snow type | Location in snow profile | Typical thickness | Signal description | Force range |
|---|---|---|---|---|
| DF | Predominantly at the surface of the profile | < 2 cm | Very low force signal | < 1 N |
| IF | Anywhere | 0.1 mm − 5 mm | Sharp singular peak, no intermediate peaks | > 1 N |
| SH | Surface of profile | < 10 mm | Tooth-like structure similar to depth hoar | 0 − 0.2 N |
| RGwp | Anywhere. Not necessarily on the surface and can sometimes be buried | 10 mm − > 50 cm | Wavy force signal, when density is around 500 kg m$^{-3}$ can also have a tooth-like structure similar to depth hoar (density of > 400 kg m$^{-3}$ is typical for Arctic wind crust) | Varying but in the 2 − 20 N range |
| DH | Often found in the middle to the bottom of the profile | Complete range | Classic teeth signal, increasing in force, then a sudden drop in force, due to hitting an air pocket | 0 − 2 N |
| DHid | Often middle-bottom of profile | Complete range | Classic teeth signal. Does not drop to 0 N like DH would | 2 − 6N (± 2 N) |
| DHwp | Very hard layer at the surface | 4 mm − 10 cm | High force signal caused by wind packed snow grains which have metamorphosed into an icy layer | 5 − 30 N |
| MFdh | Very hard layer at the snow-sea ice interface | 1 − 10 mm | High force signal caused by a metamorphosed surface scattering layer buried under the snowpack | 5 − 30 N |

## B2 Examples of snow types' SMP signals

## B3 Complementary parallel measurements

When measuring the snow properties, we had access to numerous instruments, which each proved to be beneficial when interpreting the snow grain type. For example, the near-infrared camera provided overview images of the cross-section of the

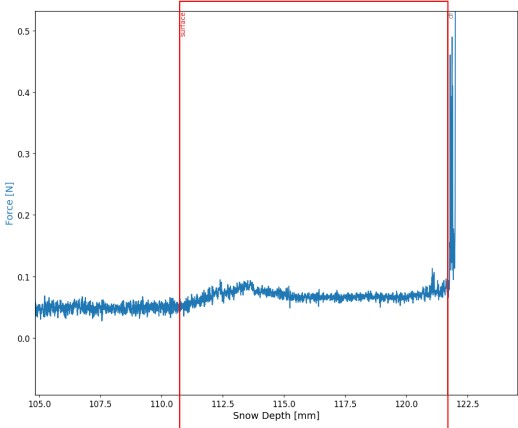

(a) A snow micro penetrometer signal showing a typical signal for decomposing and fragmented precipitation particles (DF) with a force remaining under 0.1 N between approximately 111 mm and 121 mm.

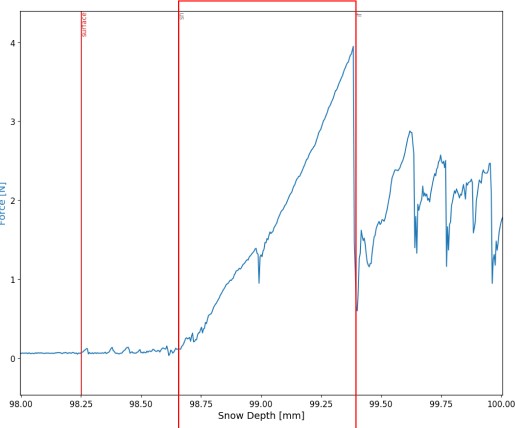

(b) A snow micro penetrometer signal showing a typical signal for ice formations (IF) with a sharp singular peak at a maximum of 4 N between approximately 98.6 mm and 99.3 mm.

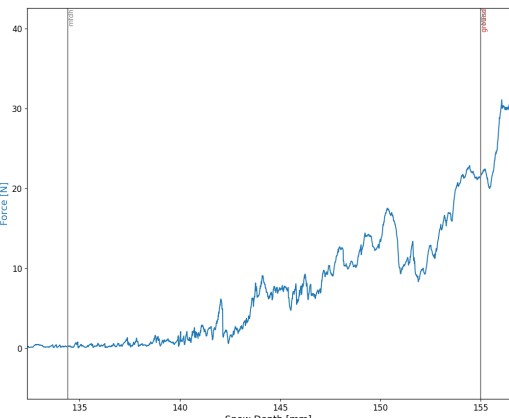

(c) A snow micro penetrometer signal showing a typical increase in force at the snow-sea ice interface. This signal is typical for a remnant surface scattering layer, named melt form, depth hoar (MFdh) in this study. This signal typically has a force range of $5 - 30$ N.

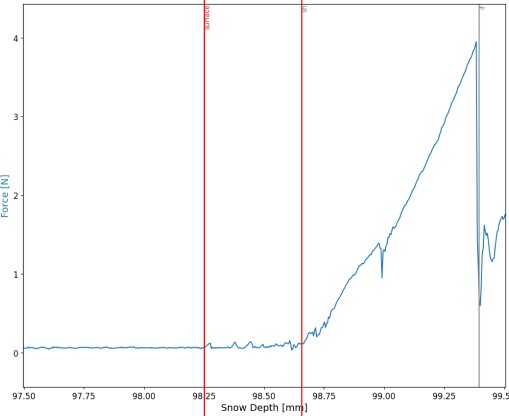

(d) A snow micro penetrometer signal showing a typical signal for surface hoar (SH) at the surface of the profile with a tooth-like structure with a low force signal.

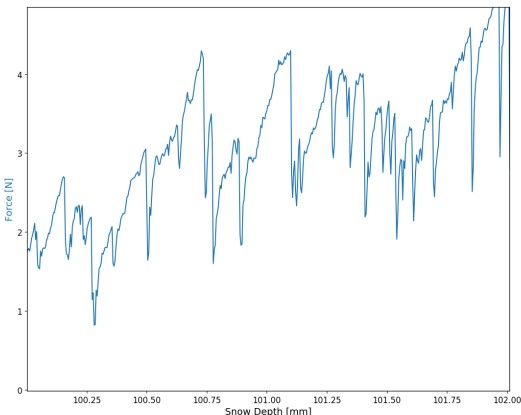

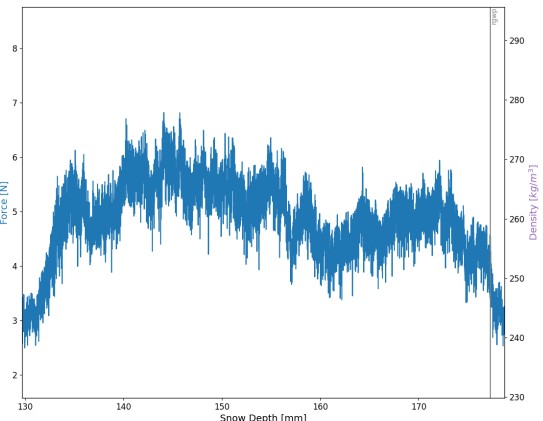

(e) A snow micro penetrometer signal showing a typical tooth-like signal for indurated depth hoar (DHid) with a force between $2 - 6$ N.

(f) A snow micro penetrometer signal showing a typical wavy force signal for rounded grains, wind packed snow (RGwp).

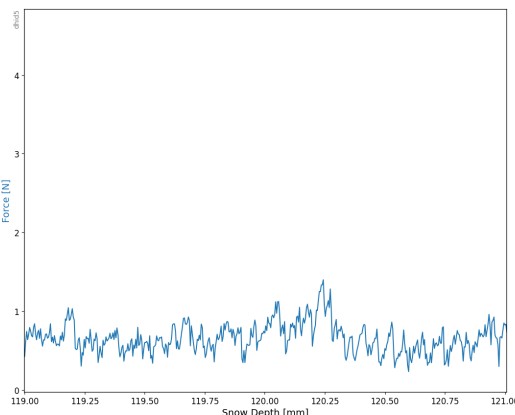

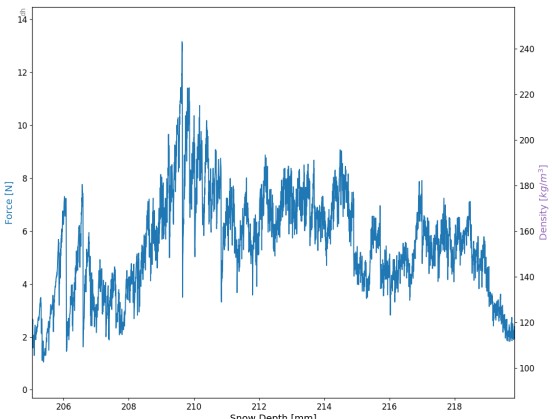

(g) A snow micro penetrometer signal showing a typical tooth-like signal for depth hoar (DH).

(h) A snow micro penetrometer signal showing a typical wavy and tooth-like signal for depth hoar, wind packed (DHwp) with a force between $5 - 30$ N at snow depths 208 mm to 215 mm.

**Figure B1.** SMP profiles with typical SMP signals for the following snow types: a) DF, b) IF, c) MFdh, d) SH, e) DHid, f) RGwp, g) DH, and h) DHwp.

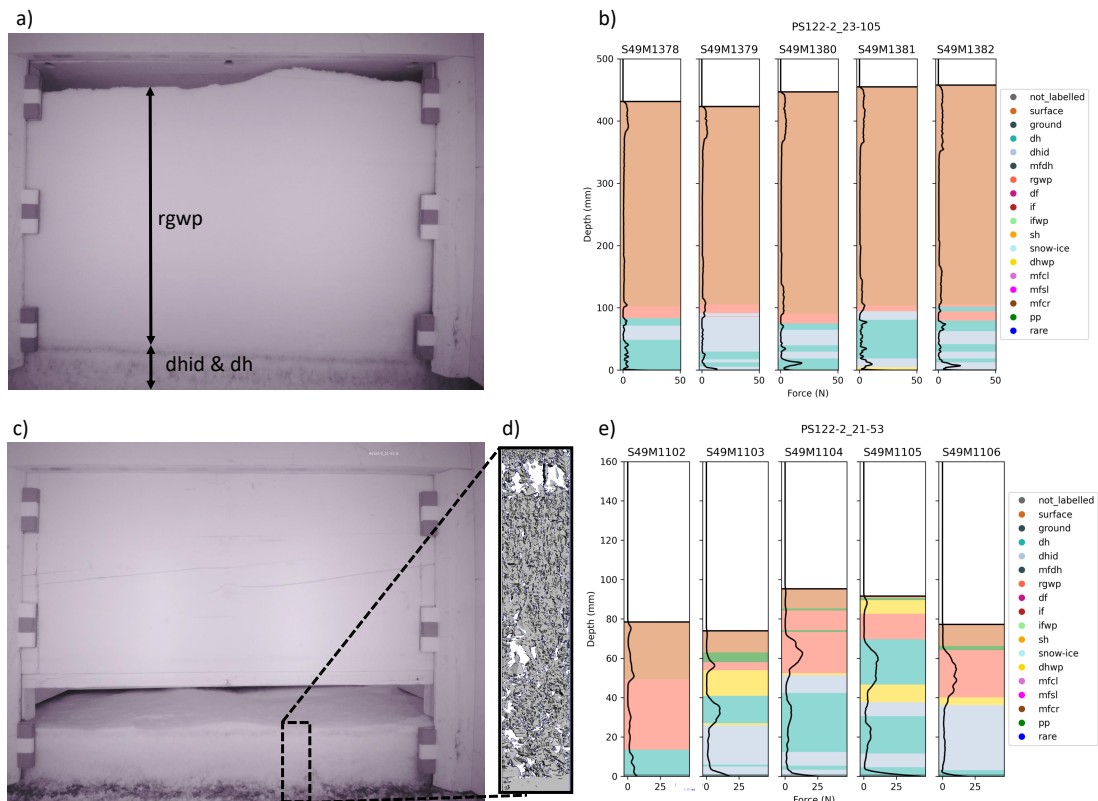

**Figure B2.** An holistic figure showing the use of a library of datasets to assist in labeling the SMP signal. a) An NIR image from the event PS122-2_23-105 giving a horizontal cross-section of the snowpack where the 5 SMP measurements in b) were taken. The rounded grain, wind-packed (rgwp), indurated depth hoar (dhid), and depth hoar (dh) regions are identified. b) Five SMP profiles measured approximately 20 cm apart in the same snowpit during event PS122-2_23-105. c) An NIR image from event PS122-2_21-53 giving a horizontal cross-section of the snowpack where the 5 SMP measurements in e) were taken. d) A 3-D reconstruction of the snow microstructure measured using micro-computer tomography. e) Five SMP profiles measured approximately 20 cm apart in the same snowpit during event PS122-2_21-53.

snowpit wall (see examples in Figures B2a and B2c), and micro computer tomography measured the snow's microstructure in
high-resolution (Figure B2d). The metadata section in the dataset by Macfarlane et al. (2021) gives additional information on how many micro-CTs and NIR images are used in parallel to each other.

## Appendix C: Features

### C1 Features included in data

Table C1 lists all features that were included in the training, validation and testing data of this study. The importance of those features depends on the specific snow type that should be classified. See Table C2 for this. For example, "Rounded Grains Wind Packed" shows a high correlation with micromechanical features such as $L$ (4 mm window), whereas "Melted Form of Depth Hoar" is mainly correlated with the force values of the SMP profile. Further feature importance analysis (ANOVA and decision tree importance) can be found online in the snowdragon GitHub repository.

### C2 Label-wise feature correlation

Table C2 shows why classification for this dataset is so hard. Some labels have lower correlations among all features, making it unclear how the right predictions can be achieved on this basis. Other more predictive features are missing, i.e. if a feature is discovered that shows a high correlation within this plot, it might boost the overall classification capabilities of the models. The figure also shows that there might be interaction effects arising since some snow types show very similar correlations (for example "Melted Form of Depth Hoar" and "Depth Hoard Wind Packed"). In summary, the label-wise feature correlation reveals the classification difficulty of the dataset and can be used to discover new predictive features.

## Appendix D: Metrics

The metrics used for validation and testing are listed and explained in Table D1. It might be helpful to familiarize oneself with a binary confusion matrix beforehand.

Intuitively speaking, accuracy expresses how many samples were predicted correctly relative to all predictions; recall expresses how many positive samples were predicted correctly relative to all positive samples; precision expresses how many positive samples were predicted correctly relative to all positive predictions; F1 score can be used to measure both recall and precision in one score; ROC is the receiver operating characteristics and plots the true positive rate versus the false positive rate; AUROC expresses, that the higher the area under the ROC curve, the clearer can the model separate between positive and negative samples; and log loss expresses how good or bad the prediction probabilities of each sample are compared to the target predictions. All these values are better the larger they are, except of the log loss, which is kept as low as possible. Some of the metrics from Table D1 cannot be computed for all models. This is the case because the AUROC and the log loss metric operate on prediction probabilities for the different classes, which not every model can provide. In these cases, the missing metric is marked with "-" in the result tables.

| Feature Name | Abbreviation | Explanation |
|---|---|---|
| distance | dist | Distance from the snowpack's surface |
| dist_ground | dist_gro | Distance from the ground |
| pos_rel | pos_rel | Relative position in the snowpack |
| gradient | gradient | Gradient (slope) of the force signal |
| mean_force | mean | Mean force signal (1 mm window) |
| mean_force_4 | mean_4 | Mean force signal (4 mm window) |
| mean_force_12 | mean_12 | Mean force signal (12 mm window) |
| var_force | var | Variance of the force signal (1 mm window) |
| var_force_4 | var_4 | Variance of the force signal (4 mm window) |
| var_force_12 | var_12 | Variance of the force signal (12 mm window) |
| max_force | max | Maximum of the force signal (1 mm window) |
| max_force_4 | max_4 | Maximum of the force signal (4 mm window) |
| max_force_12 | max_12 | Maximum of the force signal (12 mm window) |
| min_force | min | Minimum of the force signal (1 mm window) |
| min_force_4 | min_4 | Minimum of the force signal (4 mm window) |
| min_force_12 | min_12 | Minimum of the force signal (12 mm window) |
| median_force_4 | med_4 | Median of the force signal (4 mm window) |
| median_force_12 | med_12 | Median of the force signal (12 mm window) |
| delta_4 | delta_4 | Width of peaks in the force signal (4 mm window) |
| delta_12 | delta_12 | Width of peaks in the force signal (12 mm window) |
| L_4 | L_4 | Distance between neighbouring peaks in the force signal (4 mm window) |
| L_12 | L_12 | Distance between neighbouring peaks in the force signal (12 mm window) |
| lambda_4 | lambda_4 | Parameter regulating the Poisson shot noise (4 mm window) |
| lambda_12 | lambda_12 | Parameter regulating the Poisson shot noise (4 mm window) |

**Table C1.** Names and description of the features included in the training, validation and testing data.

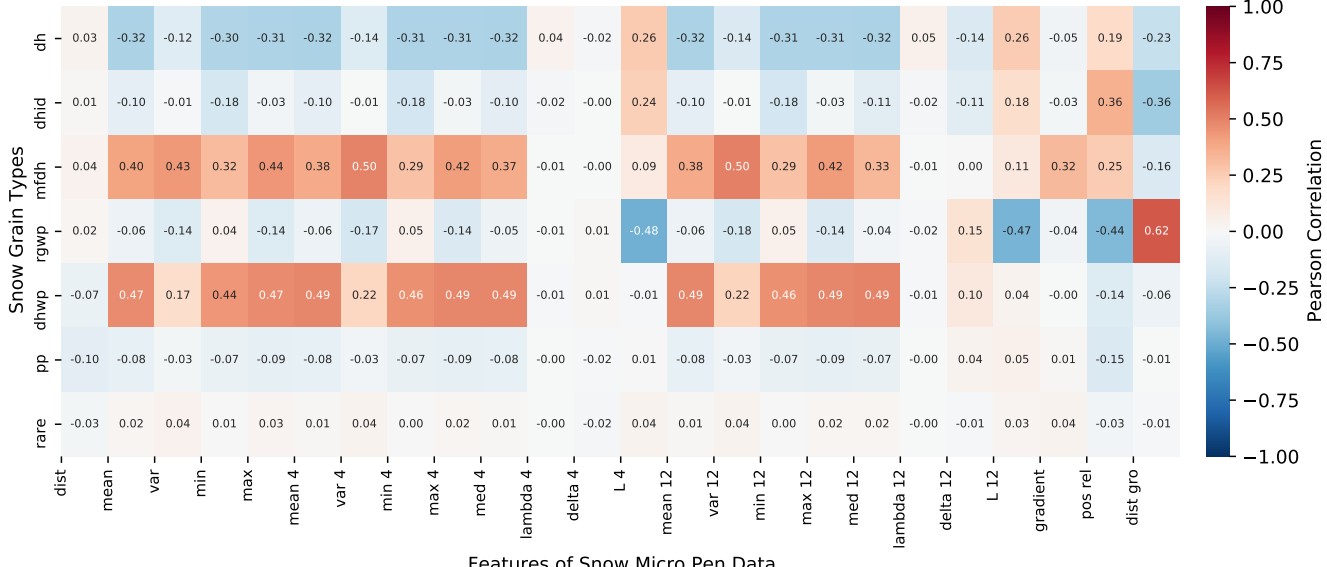

**Table C2.** Label-Feature correlation between snow types and aggregated features of the SMP profiles. The numbers in the feature names stand for the window size used during aggregation. "Depth Hoar" (dh), "Depth Hoar Indurated" (dhid), and "Rounded Grains Wind Packed" (rgwp) show some negative correlations with a subset of the features. "Melted Form of Depth Hoar" (mfdh), "Depth Hoar Wind Packed" (dhwp) and "Rounded Grains Wind Packed" (rgwp) show a strong positive correlation with at least one feature. "Precipitation Particles" (pp) does not show strong correlations with any feature, however, a correlation with distance (dist), variance, and force features was expected by experts. The low correlations could be caused by the data-preprocessing step when "Decomposed and Fragmented Precipitation Particles" were categorised as "Precipitation Particles" as well. The class "Rare" shows no correlations with the features since it consists of very different sub-classes ("Ice Formation" and "Surface Hoar").

## Appendix E: Machine specifications

The evaluation and hyperparameter tuning experiments were run on two different machines. The complete evaluation was conducted on a 64-bit system with an Ubuntu 18.04.5 (Bionic Beaver) operating system. The machine has 16 GB RAM and an Intel® Core™ i7-6700HQ CPU @ 2.60GHz × 8 (and the GPU was not used). The machine on which the first hyperparameter tuning, training, and validation experiments have been run has the following specifications: 64-bit system with an Ubuntu 20.04.1 (Focal Fossal) operating system, an Intel® Core™ i7-4510U CPU @ 2.00GHz x 4 CPU, and 12 GB RAM (and the 620 GPU was not used). Final hyperparameter tuning, training, and validation (results presented here) were run on an Azure virtual machine of the Dsv3-series, namely on a Standard_D4s_v3 [3] machine with Ubuntu 18.04 (Bionic Beaver) as an operating system, 16 GB RAM and 4 vCPUs.

---

[3]https://docs.microsoft.com/en-us/azure/virtual-machines/dv3-dsv3-series

| Metrics's Name | Formula for Binary Case | Description |
|---|---|---|
| Balanced Accuracy | $\frac{1}{2}\left(\frac{TP}{TP+FN} + \frac{TN}{TN+FP}\right)$ | Macro-average of recall scores per class. For balanced datasets, the score is equal to accuracy. |
| Weighted Recall | $\frac{TP}{(TP+FN)}$ | Calculates the recall for each class and computes the mean, weighted by the class's presence in the target data. |
| Weighted Precision | $\frac{TP}{(TP+FP)}$ | Calculates the precision for each class and computes the weighted mean, weighted by the class's presence in the target data. |
| F1 Score | $2 * \frac{\text{precision} * \text{recall}}{\text{precision} + \text{recall}}$ | Harmonic mean of precision and recall. In the multiclass case, F1 computes the class mean, weighted by the class's presence in the target data. |
| AUROC | - | Computes the area under the receiver operating characteristic curve from the prediction scores. The ROC curve plots the true positive rate versus the false positive rate. The scores are calculated for each class against all other classed (one-versus-rest) and weighted. |
| Log Loss | $-(y \cdot \log(p) + (1-y) \cdot \log(1-p))$ | Negative Log-Likelihood of a logistic model that returns prediction probabilities $p$ for the true data $y$. |

**Table D1.** List of metrics employed during validation and testing. The given formulas are only simplified versions for a binary classification case where no weighting takes place. The formula for the AUROC is not given here, since it is no one-liner and actually involves calculating an area under the ROC curve. Implementation and explanations of the metrics are from Pedregosa et al. (2011).

## Appendix F:  Model setup

The project was executed in Python 3.6 and all used packages can be found on GitHub in the "requirements.txt" file. Principle component analysis, t-SNE, k-means clustering, Gaussian Mixture Models, Bayesian Gaussian Mixture Models, random forests, SVMs, and the k-nearest neighbour algorithm were used as made available through scikit-learn by Pedregosa et al. (2011). [4] The easy ensemble for imbalanced datasets and a balanced variant of the random forest are imported from imbalanced-learn by Lemaître et al. (2017). [5] All ANN architectures were created with the help of TensorFlow (Abadi et al., 2016) [6] and Keras (Chollet et al., 2015) [7]. The attention model within the encoder-decoder network was used as provided in the keras-attention-mechanism package by CyberZHG (2020).

## Appendix G:  Pruned decision tree

---

[4]https://scikit-learn.org/stable/

[5]https://imbalanced-learn.org/stable/

[6]https://www.tensorflow.org/

[7]https://keras.io/

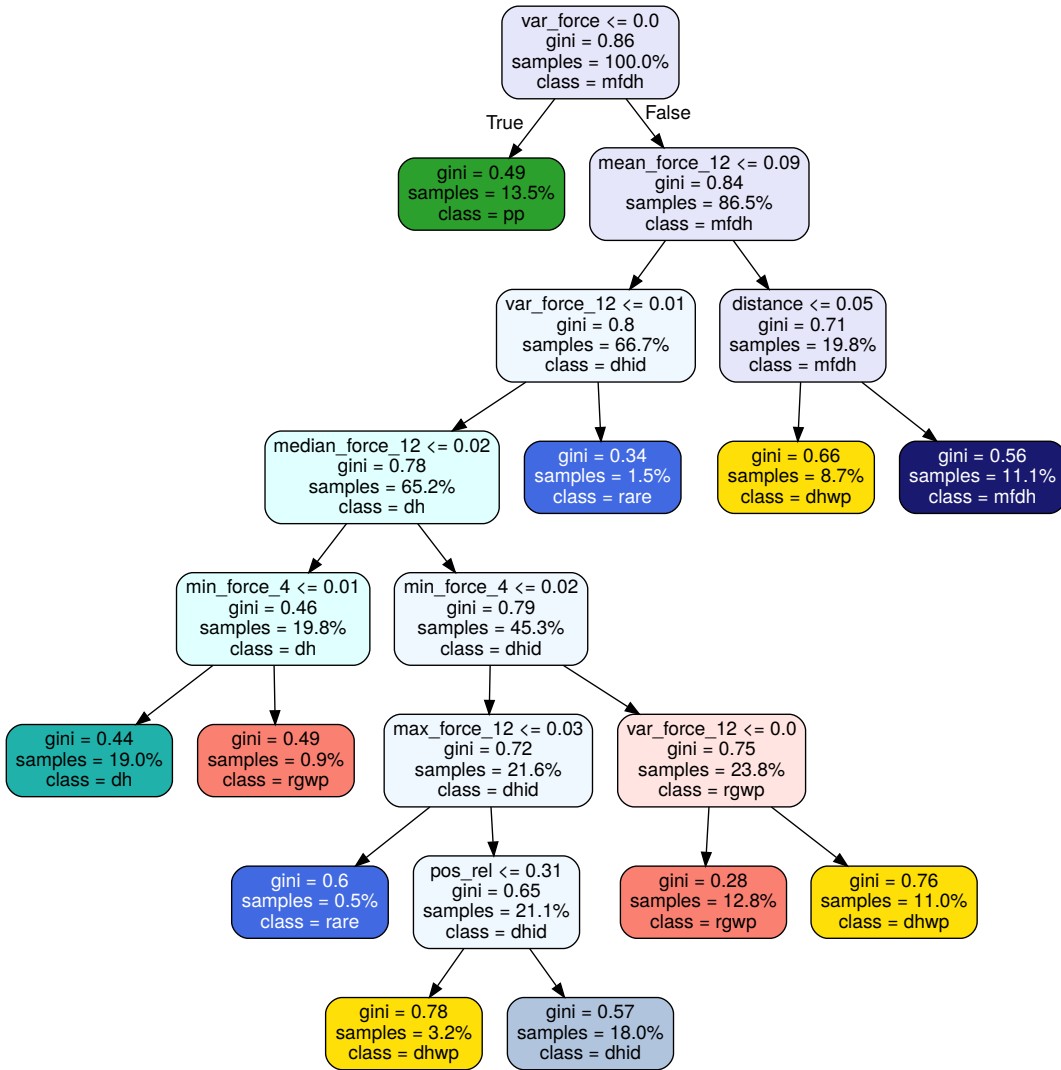

**Figure G1.** Pruned decision tree extracted from the random forest. See Appendix C1 to understand the features that the nodes represent. Decision trees encode the decision rules for predicting snow type labels. This approach helps to explain the model's decisions, a property that is often asked for by domain experts. At each leaf node, a labeling decision is made. All the other nodes encode the labeling rules that are used to classify each point. Take the root node as an example: If the variance of the force is smaller or equal to zero, the point is labeled as "Precipitation Particles". Else it has to be one of the other labels. The Gini index encodes how well separable the subsets of data points are (the bigger the number the better), and the sample's number shows how much percent of the complete data can be found in this subset.

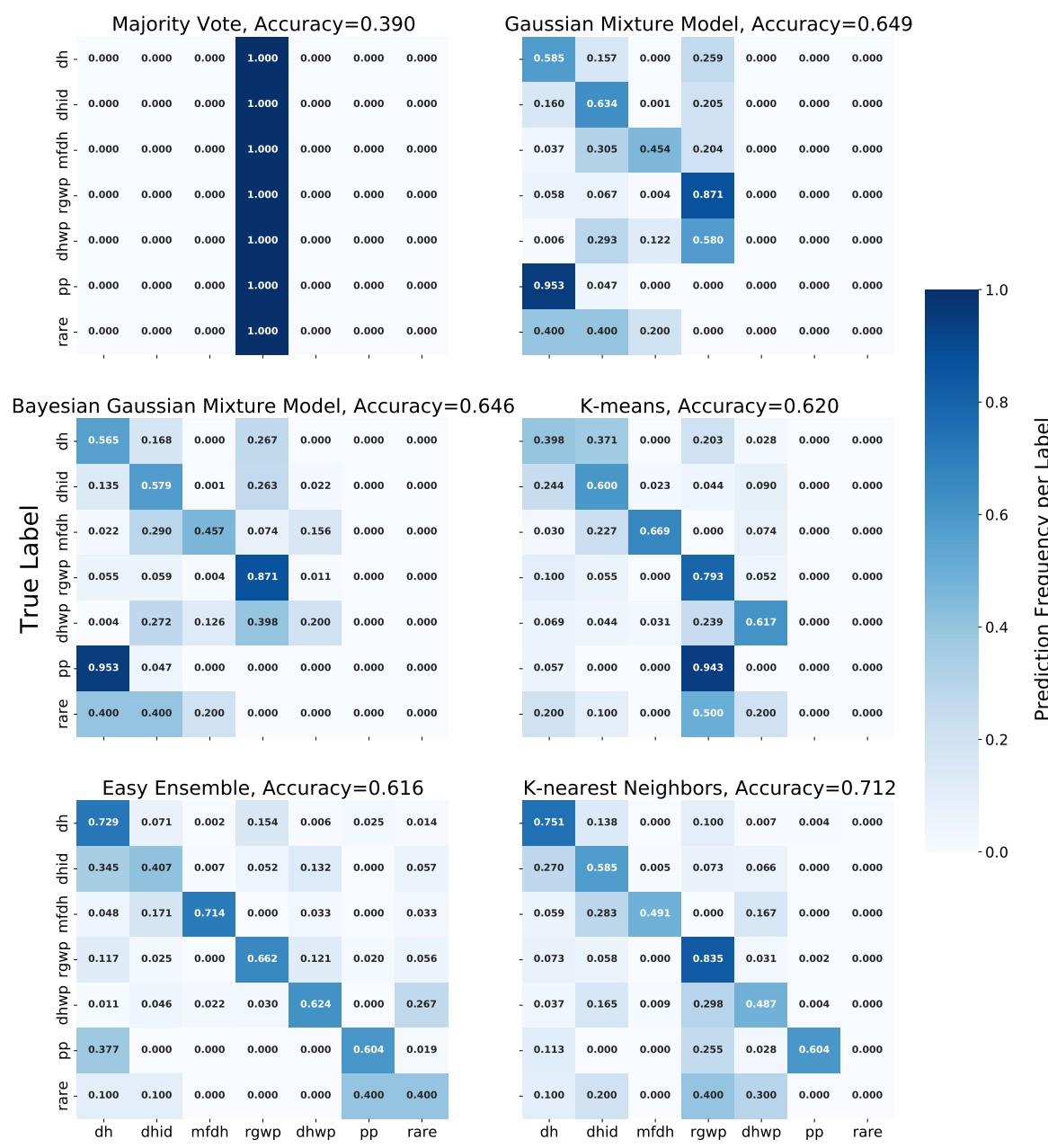

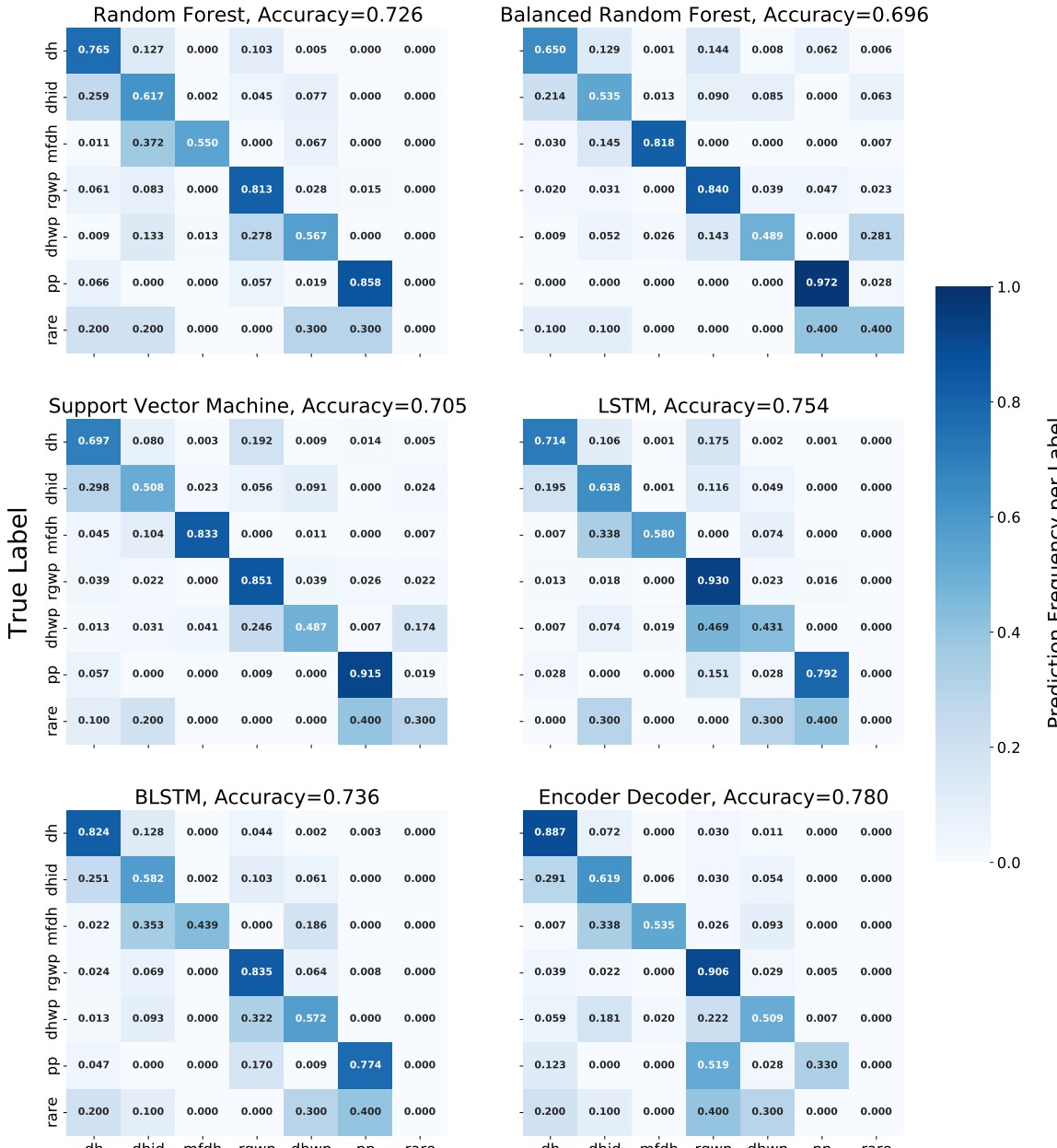

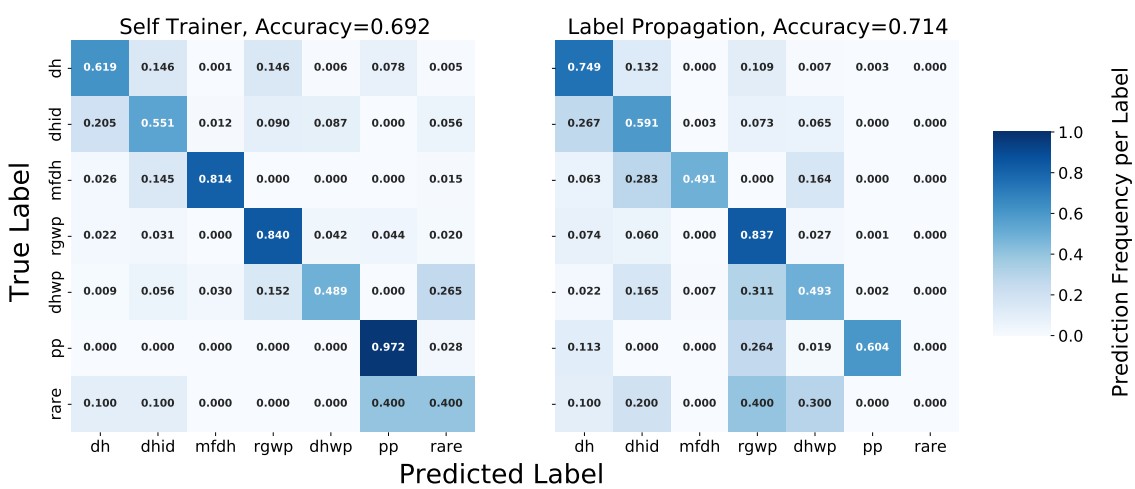

**Table H1.** Confusion matrices of all models displaying the predicted and the observed snow types. The number in each cell is the relative prediction frequency of a label within the observed class. The numbers of the diagonal (upper left to lower right) represent the prediction accuracy of each label. The stronger pronounced the diagonal and the less pronounced the upper and the lower triangles are, the better the predictions. The confusion matrices help for an in-depth analysis of the label-specific performances. This is useful when users want to choose a model that is suitable for a specific snow classification task.

*Author contributions.* ARM and MS collected and curated the data; ARM and MS labeled the data; ARM and JK preprocessed the data; JK developed the methodological framework; JK implemented, compared, tuned and validated the models; JK and VC visualized the results; JK wrote the manuscript draft; VC, ARM and MS reviewed and edited the manuscript; VC supervised the ML part of the study; MS supervised the cryospheric part of the study.

*Competing interests.* The authors declare that they have no conflict of interest.

*Acknowledgements.* This project was funded by the Swiss Polar Institute (DIRCR-2018-003), the European Union's Horizon 2020 research and innovation program projects ARICE (grant 730965) for berth fees associated with the participation of the DEARice project, the WSL Institute for Snow and Avalanche Research SLF (WSL_201812N1678). The project was additionally financed by the funds of a research training group provided by the Deutsche Forschungsgemeinschaft (DFG), Germany (GRK2340). Data used in this manuscript was produced as part of the international Multidisciplinary drifting Observatory for the Study of the Arctic Climate (MOSAiC) with the tag MOSAiC20192020. The data was collected during the Polarstern expedition AWI_PS122_00. We acknowledge the contribution of the MOSAiC-expedition (Nixdorf et al., 2021). We especially thank the crew of RV *Polarstern* (Knust, 2017) and participants of leg one to three for their help in the field. We would especially like to thank the late Joshua M. L. King for insightful discussions and comments.

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
