# Peer review of "Automatic snow type classification of Snow Micro Penetrometer profiles with machine learning algorithms"

_EGUsphere, 2022_

## Referee Comment (RC1)

[referee-annotated manuscript omitted]

---

## Author Comment (AC1)

**Author Comments**

**Main Response**

We thank both reviewers for their valuable and insightful comments. Here, we would like to summarize our main responses and changes to provide the editor with an overview. All those topics are addressed in more detail in the point by point responses for each reviewer.

Main responses:
- The SMP profile labeling has actually been fine-tuned with Micro-CTs and NIR where possible. However, we do not have such measurements for each profile. Arctic conditions (25 m/s wind, -30°C) make in situ observations such as snow grain categorization on a metal plate very difficult, and changing personnel means that those assessments would not remain consistent. Under such conditions the SMP can provide us with a large number of consistent profiles. Up-scaling a consistent labeling of those profiles is exactly the type of task that ML algorithms can tackle.
- We are only leaning towards the international classification of seasonal snow on the ground (Fierz et al., 2009). We are not using solely the classes provided there because it contains mainly alpine snow types and does not cover all typical Arctic snow types. However, we still use snow classes in general - instead of abandoning this concept completely - since we strive to use a common language of the cryospheric community.
- The work presented here cannot and does not aim to remove the subjectivity of snow grain type classification. The models provided here generate a labeling that is consistent with the training data. As long as the training data is subjective, the predicted profiles will remain subjective. The subjectivity of snow grain type classification is not a problem confined to the SMP - in situ grain type classification on metal plates, snow pit stratigraphy analysis, NIR stratigraphy analysis, etc. - all of them are of subjective nature.

- The work presented here is a methodological contribution, evaluated on numerical metrics. We also evaluated from a qualitative perspective if the predictions on an out-of-distribution dataset behave in a way that a practitioner (who labeled the training data!) would: 1) accept those predictions, 2) find them consistent with their own labeling, 3) and subsequently work with these predictions. We find this qualitative assessment important since it decides if the tools provided here can be used in practice.
- The main objective of this study is to provide a way to up-scale manual SMP labeling and make the life of practitioners easier this way. We want to simplify the complex and longish procedure of SMP classification by providing ML algorithms that can reproduce a trained labeling pattern on new profiles. This way larger datasets can be processed and analyzed.

Main changes:
- We are going to provide a detailed description of the labeling process (partially in the manuscript, in-depth in the supplementary materials) and the environmental context in which the data has been collected.
- We will provide a description of the different micromechanical properties of different snow grain types. We will describe how they influence the SMP signals and thus the overall classification through ML models.
- We would like to sharpen our manuscript regarding the main objective of this study.
- We will provide a more detailed description of the included input features, the metrics, and some ML termini.
- We also want to change our wording at several points, e.g. we will not use the word "ground truth" anymore, use the wording "depth-dependent" information, we will make clear that we are only leaning towards Fierz et al. (2009), and we will delete the "segmentation" part from our title.
- We will add a detailed user guide to the supplementary material.
- You can find more changes we plan to make in the point by point responses.

We were pleased to hear that we provide important work to test numerous machine learning models on this task, and that the computer science aspect of our study was complete.

We would be very glad if we could present a revised version of our manuscript that includes those changes and several more as suggested by our reviewers. We thank everyone for their time and support!

**Reviewer 1**

**General response**

Thank you very much for your in-depth feedback and for providing us with such helpful comments. All of your comments are very much appreciated and have helped us to improve the manuscript. To summarize the most important responses: The profiles have been labeled with additional in-situ observations at hand (Micro-CT and NIR) – we added more information on the complete labeling process in the manuscript and provide a comprehensive overview in the additional complementary material. We will also include a more detailed discussion about the micro-mechanical properties of the different snow types and the relation between classification difficulty and micro-mechanical properties. In our responses you can find an explanation about the "qualitative nature" of the validation process, why we find it important to include such an evaluation and why this study cannot solve the subjectivity of snow grain classification in general. We hope that you find all your other suggestions addressed in the below responses. We found them all very helpful and will include them in our revised manuscript. Thank you for your time and helping us improve the manuscript significantly.

**Point-by-point responses (following the order of the comments)**

- *[2] consider rewording as, "at submillimetre intervals in snow depth."*

  - Accepted, will be reworded.

- *[12] The software "snowdragon" has not been previously explained. Incorporate a sentence within the abstract to give the reader context to this software, as to avoid confusion.*

  - Agreed, thank you so much. The following sentence will be added: "The findings presented will facilitate and accelerate snow type identification through SMP profiles. Anyone can access the tools and models needed to automate snow type identification via the software repository ``snowdragon''. "

- *[31] I caution against using this language, as proposing the replacement of trained scientific specialists with a "blackbox" software raises philosophical discussion which is beyond the scope of this manuscript.*

  - Thank you for pointing this out! We used to have a larger paragraph here explaining how remote sensing scientists working on just one project might prefer using a software instead of learning to categorize snow types for just one project. Now, this sentence only states the replacement of trained scientists, which is absolutely not what we wanted to communicate, so thank you for bringing our

attention to this sentence.

- ○ We will rephrase the sentence:
  "...(3) support interdisciplinary scientists who are unfamiliar with snow type categorization…"

- *[32] Consider a segue between the previous and the next using an introductory sentence such as, "Snow type classification has previously been accomplished using supervised ML algorithms."*

  - ○ Agreed, the paragraph starts very abruptly. New sentence:
    "Several previous works have addressed the task of automatically classifying snow grain types with multivariate statistics or machine learning algorithms."

- *[37] Consider joining the following three paragraphs into one, as they are a bit meager to stand alone.*

  - ○ Agreed. We will join the paragraphs and add the following conclusion sentence at the end:
    "Thus, previous work showed that supervised machine learning algorithms are a promising pathway to automatic snow grain categorization."

- *[52] I think the novelty of your approach is quite apparent and this disclaimer does not add value for the reader.*

  - ○ Accepted. The disclaimer will be removed.

- *[general] This term "ground truth" is misleading, as the interpreted profiles were not validated by an exhumed pit. Consider referring to these as 'Labeled' throughout.*

  - ○ We completely agree that ground truth is misleading, as there is no such thing as ground truth. We will adapt this throughout the manuscript. We will also update Figure 3 accordingly.

- *[75 – 76] Without validating the SMP force profile labeling, how can you be confident in the interpretation? Some discussion is given on this point later, but I think an additional sentence or two, which clearly states that no in-situ comparisons of grain-card type cryptography, micro-CT scanning, NIR photography, or SSA measurements were collected with the SMP observations. You must convince the reader here that your entire ML methodology which relies on these labels is still valid. A statement on the confidence and expected uncertainty in these interpretations is needed in the methodology section. And a justification as to why no corresponding validation measurements were collected, even if the explanation is that it is too cold or windy on the sea ice to bother with these*

*observations, needs to be provided.*

- ○ Thank you for that valuable comment. We will add a paragraph explaining what is going on here and pointing out that Micro-CT data has been used to fine-tune the labeling where possible. We will also add an appendix that explains the complete labeling process in more detail and makes it hopefully more transparent to the reader.
- ○ The paragraph that will be added:
  "The labeling was conducted by a snow expert and is based on the properties of the force signal (magnitude, frequency, and gradient) and the signature of the SMP-signal \citep{schneebeli1999measuring}. Micro-CT samples and NIR photography were used to fine-tune this process by validating the grain types identified from the force signal. However, these additional measurements are not available for each SMP profile for the following reasons: 1) Time constraints, i.e., only a few hours were available to perform all measurements within one snow pit; 2) harsh conditions on Arctic sea ice make snow pit measurements challenging."

  "Throughout the expedition, there were different operators conducting the snowpit measurements. As a result, stratigraphy analysis and in situ snow grain classification from snow pits would not be continuous since they vary from person to person. We reduce the subjectivity of in situ snow grain classification, which would introduce variability in the dataset. Instead, we use one person to create the training dataset. This reduces operator biases. The SMP is able to provide profiles fast, without physical labor, and independently from the person who measures them. The labeling procedure that was conducted on the collected SMP profiles is described in more detail in Appendix \ref{app:labeling}."

- *[77] What is the quantifiable difference between two expert interpretations of the snow types for the SMP signal? How can one deduce a particular snow type, and distinguish it accurately, from a qualitative look at the SMP signal?*

  - ○ We will address those issues in the additional supplementary material.

- *[82] For reproducibility, explicitly state which features were included. I can only assume that the mean, variance, max, and min force values of sliding windows and some unnamed mechanical properties derived from the shot noise method were included for the analysis.*

  *How important are the micromechanical features in classification compared to the force penetration profiles?*

  - ○ Agreed, we cut too much information here – thank you for pointing us to this. We will adapt the relevant paragraph and add a Table in the Appendix that lists all features included in the data. Regarding the importance of the micromechanical

features: It depends a bit – we did ANOVA and decision tree feature extraction, which ordered the importance of each feature. Taking ANOVA results, we see that the micromechanical features are not as important, however, the decision tree importance does actually estimate L (4 mm window) as the 4th most important feature. We added a table at the end of this document on that matter. We found it more helpful to look at the feature importance for each grain type separately because different features are more or less important for each grain type. We will reorder our Appendix, add a few lines, and also refer in the main text to the feature correlation heatmap on that matter now. In the heatmap, you can see that e.g., for rounded grains wind packed, the micromechanical features are very important, whereas for melted form of depth hoar, the force values are much more critical.

- ○ Adapted paragraph:
  "For each SMP profile, we replaced negative force values with 0, summarized the signal into bins (1 mm), and added mean, variance, maximum, and minimum force values for those bins. Those values were also determined for a 4 mm and 12 mm moving window. Moreover, \citet{lowe2012poisson}' Poisson shot noise was used to extract $\delta$, $f$, $L$, and the median force value for a 4 and 12 mm window. We added further depth-dependent information by including for each data point the distance from the ground and position within the snowpack. Refer to Table \ref{tab:features} in Appendix \ref{app:features} for an overview of all features used for each SMP profile, and to Table \ref{tab:feature_corr} to see the feature importance for each grain type."

- *[84, 90 etc.] Because the SMP is measuring force as a function of depth, depth-dependent seems like a more clear explanation. It took me a while, but I understand time-dependent as the snowpack history, which accumulates in time. [and all related comments]*

  - ○ Thank you for this feedback – the main author is used to thinking in machine learning terms and framing this as "time-dependent" information (because each data point has been measured sequentially after each other == "time-dependent"). Depth-dependent does capture this really much better, and we will adapt this everywhere.

- *[89] Although for this classification purpose, it may be convenient to compile these snow types into the rare category, ice formations and surface hoar have widely different mechanical properties. Therefore we can almost expect the classification to perform poorly on the rare class. I appreciate the brief discussion on the value of separating these snow types for avalanche hazard assessment, but if you were to take an SMP profile in the rocky mountains of the western US this year, buried surface hoar would be a widely extensive class, not rare.*

  *Drafting some discussion regarding the snow mechanics as a classification device,*

*rather than the rare appearance of this snow type in a dataset is lacking from this work.*

- ○ We are completely aware that the classification performance on the class "Rare" will be bad due to the different mechanical properties. We did this to simplify the evaluation of the models and in order to not skew the balanced metrics too much: If we have many rare classes where it is almost impossible to achieve good performance (not enough data for ML), this will lower the overall accuracy heavily since each class is weighted not according to occurrences but according to the overall number of classes. Hence, we had the feeling that the overall performance of the models is easier to evaluate for everyone if we summarize those very rare grain types into one class. We also could have dropped them completely as commonly done in previous work, however, we still wanted to include those occurrences because it was important to us to show how models perform on a real-world dataset – and each dataset will usually have some rare classes, and in our opinion, it would be a loss to force practitioners to drop those profiles.

- ○ Regarding the notion of which grain types are "rare" – we completely agree that this is heavily dependent on the dataset at hand, and we invite everyone to retrain snowdragon for their specific datasets. And we highly encourage summing up other grain types to "rare" in those contexts. We will include a more detailed user guideline to make this more transparent. We will also adapt the text to make clear that the notion of "rareness" only applies to the MOSAiC dataset and is not meant as a general categorization.

- ○ Adapted text:
  "The few occurring ``Ice Formations'' and ``Surface Hoar'' instances in the MOSAiC dataset are summarized in the class ``Rare''. While a high classification performance cannot be expected for the rare classes, we still include them to show how the models perform on a ``real-world dataset'' that in most cases will also include classes with few occurrences."

- ○ Regarding your comment on snow mechanics as a classification device, we will answer this later in our response.

- ● *[93] Balance and imbalance are ML jargon terms that could be more clearly defined to improve the readability and interpretation of the results.*

  - ○ Yes, that is true, thank you. This will be adapted in the following way:
    "The resulting dataset has the following properties: (1) There are multiple, noisy, and overlapping classes. (2) There is a between-class imbalance, i.e. some grain types occur much more frequently than others. (3) There is a within-class imbalance, i.e., some grain classes contain different sub-grain classes, but some

of them are more frequent than others."

- *Minor corrections regarding the abbreviations of AUROC etc.*

  - All accepted, will be changed accordingly.

- *Minor corrections regarding "generalized data"*

  - We prefer naming this specifically "out-of-distribution" data. Generalized data transports the message that if we train a model on this "generalized data", it can actually generalize to anything. If we use "out-of-distribution" data, we can test the generalization capabilities of a model, but it does not entail that the model can generalize to anything. A model trained on "generalized" data sounds like it could generalize to anything.

  - We still will make the following adaptations:
    "(3) The generalization capability is tested by running the best-performing model on 100 random profiles from different parts of MOSAiC winter data. These profiles are outside the distribution of the training, validation, and testing data, and we refer to them as ``out-of-distribution profiles``. Here, the ``out-of-distribution'' profiles contain the same classes as the training data, so the model still has a chance to predict the correct labels."

- *Other linguistic suggestions on Page 6*

  - All accepted, will be changed accordingly.

- *[167] This is an assumption, as the generalized data are not validated. It is better to write, "however, if the general data contains the same classes as the training data…"*

  - Yes, we made sure to choose an out-of-distribution dataset that contains only labels known to the ML models. We will adapt the text to make clear that specifically in our case, we made sure that the out-of-distribution dataset from which we draw the profiles contains only labels known to our models. (See paragraph mentioned above).

- *[162] Expand this section to include more detailed description of how accuracy is calculated, how balanced accuracy differs/what information is conveyed by this metric, how weighted precision is used as an uncertainty metric. This is explained for AUROC, and should be explained for the other metrics to give a general reader context to the evaluation metrics. F1 score is undefined*

  - We agree that a more detailed description of the metrics is important for all the readers who do not work with those metrics on a daily basis. We found it most

helpful to add another appendix to this end so we are able to provide both formulas, definitions, and intuitive explanations for all metrics. We are also defining the F1 score now.

- *[171] Consider joining this section with the previous Section on Evaluation, as the section is quite short to stand alone.*

  - In machine learning papers, it is common to separate the evaluation and the experimental setup, so the reader can look up the experimental setup immediately to check if they have the means and resources to reproduce the experiments. We understand, though, that the sections were quite short to stand alone. We are going to add some more details to the experimental setup, and with the planned changes of the evaluation section, we hope the paragraphs will be able to adequately stand alone.

- *[175] Define/Explain hyperparameter tuning*

  - Adapted paragraph:
    "Hyperparameter tuning is the process of searching for the optimal internal learning settings of an ML model. Hyperparameters control the learning process of the models, whereas parameters are learned by the model. The tuning is performed on the validation data and the hyperparameters that achieve the highest performance for their model chosen for subsequent model evaluation. Here, hyperparameter tuning was applied moderately and with a simple grid search. All tuning results can be found in the GitHub repository. Specifications of the machine on which the experiments were run can be found in Appendix \ref{app:machine_specs}, and descriptions of the model setup can be found in Appendix \ref{app:model_setup}."

- *[Figure 3:] This example between 150 - 200 mm on the Medium depth profile gives me a bit of pause. I would agree with the LSTM model here, which defines this fairly obvious series of layers. If I were to interpret this data, I would have a very difficult time discerning the snow type of this layer. I am not calling into question here, the interpreter's decision, but without validation I struggle with confidently recognizing such layering as a homogeneous snow type.*

  - We agree that without validation, it is arguable if that layering is a homogeneous snow type. The two peaks in the "medium" profile could (probably) be more traditionally classified as "wind crust", however, we did not include this class. It is interesting though that all three ML classifiers classify the peaks differently, hinting to a larger uncertainty in these predictions.
  - Your observation is an example of how ML models can support practitioners in their analysis: Essentially, the LSTM model tells the user: "Inferring from how you labeled your other profiles (training data), I would suggest the following series of

layers". Throughout the process of this study, we observed that the LSTM model can actually help to discover inconsistencies in the labeled data or human mistakes.

- *[217] This result leans into the hypothesis that it is mechanical properties that are more differentiable between classification than the count of appearances. Precip particles are a unique class distinguished by the relative lack of bonding among fresh snow.*

    - Thank you for pointing this out – we will now discuss this in more detail in the discussion section than we did before.
    - Mechanical properties of the snow influence the penetration force signal both in magnitude and characteristics of the signal. By evaluating the signal we have taken into account the mechanical properties of the snow layer. This will be explained further in the additional supplementary material.

- *[219] Explaining exactly which characteristics would significantly increase the significance of this work.*

    - The characteristics of each snow grain classification will be outlined in the new supplementary material.

- *[224 – 225] Possibly because "Rare" is comprised of mechanically different snow types, while "Precip Parts" is comprised of mechanically similar snow. Analysis of the micromechanical properties that are inverted for via the shot noise approach would increase the value of this and subsequent discussion.*

    - See response for [217].

- *[general] A general response to all comments regarding the micromechanical properties of the snow types and how they can be used as a classification device.*

    - We will give a physical reasoning why these different snow types differ in their micromechanical properties and how these properties develop through metamorphic processes.
    - That they can be used as a classification device is already entailed since the micromechanical properties create different types of SMP signals, and we classify exactly those signals with our models.
    - For us, it was a general problem that the International Classification does not represent very well snow types occurring on arctic sea ice.
    - We will add parts of this - where appropriate - in the discussion and provide more detailed descriptions in the supplementary material.

- *[254] Analysis of the spatial variability of snow class composition derived from ML prediction would be a valuable contribution which could justify this claim. In lieu of such*

*analysis, draw from the literature to better quantify the length scales of variability for snow on sea ice.*

- ○ We believe spatial variability analysis of this dataset is beyond the scope of this paper, but we wish to conduct this analysis in the near future.

- *"grouping" instead of "island"*

  - ○ Accepted, will be changed accordingly.

- [310-311] This hypothesis is a bit unrefined. All snow types are transformative and related to one another through metamorphic processes. Please clarify this statement with more physically-based reasoning. As it is written, precipitation particles should be equally non-separable because all snow has metamorphosed from fresh precip. Include indurated hoar in this discussion. What about the mechanical properties of these snow types is similar (or different) which may cause difficulties in their classification?

  - ○ Thank you for your comment, we will adapt this paragraph and discuss the metamorphic processes and the mechanical properties of these snow types in more detail as mentioned in our previous responses. Once again, the mechanical properties are the underlying driver for the classification, even when we speak about metamorphism. If two data points are very close to each other in terms of metamorphism, it entails that their micromechanical properties are close to each other. The transformation between the snow types means a transformation of their micromechanical properties. And similarly, as you suggest that some snow types have mechanical properties that are more similar to each other, we suggest here that there are metamorphism states that are more similar to each other than others.

- [314] Despite the evidence supporting that 80% accuracy appears to be a contemporary maximum for classification accuracy, the claim that 100% accuracy is virtually impossible is not justified. This language presents the notion that further advancement in this field of science cannot be achieved, and I caution against communicating in a way that paints you into a corner based on this opinion.

  - ○ In the ML community, it is quite normal to assume that 100% accuracy cannot be achieved – such a model would just be overfitting. Of course, there is a lot of space left between 80% and 100%, and we do absolutely think that further improvement is possible. But we also want to communicate that one cannot aim for such high accuracy as the ML community is usually aiming for, e.g., on the MNIST datasets. We found it important to communicate that a relatively low accuracy of 80% (for the ML community low, given their "perfect" datasets) is still a lot in a setting where classes are not clearly separable from each other.

- ○ Nevertheless, we will adapt the wording since it was apparently a bit too extreme:
  "it is currently impossible to reach 100% [...]" instead of "it is virtually impossible to reach 100% classification accuracy on every snow type since some snow types will always lie between two categories".

- *[319] This type of uncertainty can be reduced with in-situ observations of snow type through methods of crystallography etc. The labeling process is not intrinsic to the SMP analysis. The design of the experiment presented relies solely on interpretation of SMP profiles, and this choice should be discussed here. Any quantifiable uncertainty that was learned through repeated expert labeling should also be included to shed light on the value of ~80% accurate snow-type classification.*

  - ○ Yes, we agree with you, thank you for bringing this up. We will adapt this part in the following way:
    "The uncertainty during labeling is an inherent problem of SMP analysis: The annotation of SMP profiles is subjective, meaning that two different snow experts may produce two different labeled and segmented profiles for the exact same measurements \citep{herla2021snow}. This intrinsic uncertainty can be partially mitigated by supplying additional in-situ observations of the snowpack, e.g., through methods of crystallography or Micro-CT measurements. But even with additional observational data, experts might provide different annotations of the same profile. Both experts might agree that both labeled profiles are valid analyses of the same profile though. In conclusion, the model's performances cannot only be measured in terms of accuracy because models with low accuracy might still produce sensible, directly usable predictions."

  - ○ We are bringing up the topic of quantifiable uncertainty as well now in the discussion – thank you for coming up with this idea:
    "As previously discussed, the uncertainty of the expert labeling is a general limitation of this particular study. While this uncertainty might be partially mitigated further by using a dataset for which many additional in-situ observations exist, it would still remain an issue. One approach for future work would be to quantify the uncertainty that is inflicted upon the labeled profiles. Subsequently, a machine learning model could be trained to classify not only grain types but provide a \textit{probabilistic} classification."

- *[323] fix the spacing surrounding the clause*

  - ○ Accepted, will be changed accordingly.

- *[351] This sentence adds little to the discussion. It seems as though this result is "completely unclear" because such analysis has not been completed. One possibility to explore, in lieu of this large and complete data set, would be to train an LSTM model on*

*SMP data collected from a different time and place with similar snow types, predict the classification on the winter mosaic data, and validate the results on the labeled profiles. While this analysis would not clarify if one large dataset driven model would be beneficial, it would give clarity on the spatio-temporal transferability of this technique.*

- ○ Thank you. We will adapt our paragraph following your suggestion:
  "In theory, a large enough model trained on a large enough dataset could be able to produce direct predictions for any SMP users. Thus, it would be interesting to train an ML model on a generalized dataset and validate its' performance on the specialized MOSAiC SMP dataset. This would shed new light on the spatio-temporal transferability of the ML models presented here."

- *[367] The qualitative sell of this method is my largest grievance with this work.*

  - ○ We would like to point out that the work provided here is really a methodological paper that compares different machine learning algorithms for classify SMP profiles. The qualitative aspect of snow grain classification is a general issue and well known in the snow community. We are here introducing a new method to classify snow grain types but we are not suggesting that we are removing the subjectability of this research field. We are providing tools that could be used as an alternative and in the future, when the conditions in the field allow or need such a tool.
  - ○ Your comment seems to target the fact that we have not only used numerical metrics to analyze the performance of the models but also qualitative measures, namely the feedback of snow experts. (Snow expert provides labeled profiles and checks if the models create profiles that are consistent with his/her/their labeling). We do this, because this is considered an important measurement within the field of applied machine learning. While numerical measurements such as accuracy and precision might give the impression that a task is well tackled, it often can happen that domain experts criticize the predictions of the model based on features that have not been captured by the numerical metrics. Thus, it is an important addition for us, to show that domain experts have looked at the predictions of the models and deemed them as "usuable" in the field, because this is a qualitative measure often missing in ML studies. We want to make sure that the impact of this study is not purely theoretically and this is what we want to express with this sentence in our conclusion.
  - ○ An alternative qualitative measurement would be to compare NIR profiles with the ML-classified profiles. One could argue that such a comparison has a higher confidence. However, it still remains a qualitative measurement and will not become more objective. If you want to create labels that are very much aligned to NIR profiles, you can label the training data while using NIR photographs for each profile. The models, e.g. the LSTM, will then be able to create predictions that are particularly near to NIR photography because they were trained to do so. How well they are sticking to the desired prediction can only be evaluated

qualitatively, since we have no ultimate objective ground truth data.

- *[373] "Snowdragon" was mentioned in the abstract without definition, and was largely left out of the manuscript. I think an introduction to snowdragon is needed in the abstract, and the recap of the snow dragon repository needs to be explained in a more straightforward manner in the conclusion.*

    - Yes, thank you – somehow, this slipped our attention, we try introducing snowdragon properly now:

        - Abstract:

          "The findings presented will facilitate and accelerate snow type identification through SMP profiles. Anyone can access the tools and models needed to automate snow type identification via the software repository ``snowdragon". Overall, snowdragon creates a link between traditional snow classification and high-resolution force-depth profiles. With such a tool, traditional snow profile observations can be compared to SMP profiles."

        - Contributions:

          *"[…] The snowdragon repository that provides the tools to automate SMP labeling"*

        - Conclusion:
          We will remove it here at the point where you left the comment. This paragraph was more about pointing out the general implications for the field and was not supposed to be so much about snowdragon itself. It will still be mentioned in the conclusion as a "repository", but we will separate between study and repository more strongly in the conclusion.

        - Appendix:
          Proper user guidelines on how to use snowdragon

- *[378] delete "already today"*

    - Accepted, will be changed accordingly.

- *[380] ~ Glittering Generalities ~ Describe how automated snow-type classification is essential for understanding patterns of climate change. The context as to how this work would mitigate climate change impacts is not described. Consider revising this language to more clearly state what is accomplished by this work and how it is beneficial.*

- ○ Yes, agreed, the last sentence is indeed a bit too broad.

- ○ New suggestion:
  "Snowdragon enables the analysis of the SMP MOSAiC dataset, a dataset containing detailed information about snow on Arctic's sea ice. In times of climate change, this information is crucial: We need to understand the state of the sea ice in order to understand in which state the Arctic system is. For the first time, MOSAiC enables the scientific community to have access to such a detailed and large dataset. And snowdragon is one example of how ML can help us to access the knowledge behind all the data."

**Reviewer 2**

**General response**

Thank you for your comments and pointing out which parts rised unclarities or further questions for you. We hope that our additional explanations provided below make the main objective of the paper clearer for you. We also hope that our suggested changes will make sure that those concerns do not arise anymore for the future reader.

The most important changes to address your comments at a glance:

- We will include a detailed description of the labeling process and explain the context of the labeling in more detail.
- We will include a comprehensive usage guide for the snowdragon repository.
- We will extend the description of the previous work.
- We will include a more detailed description of the input data ("features").
- We will make it transparent early on that we are only leaning to the international classification of seasonal snow on the ground and why.
- We will change over-ambitious wording.
- We will change the title to communicate the "classification" part more strongly than the "segmentation" part.

Thank you for your time and your feedback.

**The objective of the work**

- "*I am not fully getting the final objective of the paper. What is the scientific question we want to address by automatically reproducing grain shape class inferred from penetration profiles based on undescribed expert analysis?*"

  - Classifying snow grain types is important in microwave remote sensing of sea ice. As recently conjectured in Picard et al. (2022)[1]. Arctic depth hoar should have a significant influence on the microwave backscatter of snow on sea ice, when compared to other snow grain types. The snow measurements conducted on the MOSAiC expedition constitute a missing puzzle piece in understanding microstructural controls of microwave scattering. Developing methods to understand the spread of depth hoar across the Arctic is just one of numerous applications of this work. Current methods to classify snow grain types rely on in situ snowpit classifications with a magnification lens and a subjective operator. But this wasn't feasible in the harsh conditions during the high-Arctic winter. We could not assess a grain type on a metal plate in 25 m/s winds and the
* * *
[1] Picard, G., Löwe, H., Domine, F., Arnaud, L., Larue, F., Favier, V., Le Meur, E., Lefebvre, E., Savarino, J. & Royer, A. The microwave snow grain size: a new concept to predict satellite observations over snow-covered regions (2022)

temperatures close to -30 °C. This study suggests an alternative approach to grain classification. We took quick and easy SMP measurements in the field and we introduced a new method of classifying grain types through the SMP signal. Details are added in the supplementary material. However, we found this process to be extremely time consuming, and to up-scale these measurements we introduce a machine learning approach. This project therefore acts as a stepping stone for further analysis of snow classification, not just on sea ice, but also in alpine regions and other cryospheric disciplines.

○ We provide here a comparison of machine learning algorithms to classify SMP profiles by approximating the labels and segmentations from a set of previously labeled profiles. We want to up-scale ("automate") the common process of an SMP practitioner who has to label each single SMP profile manually otherwise.

○ The scientific question and main objective addressed here is to find ML algorithms that are useful for a generalization of labeled SMP profiles.

● *By definition, the grain shape class or snow types (Fierz et al., 2009) is related to the shape of the grains and is traditionally derived from the observation of single grains on a crystal card with a magnification lens. This measurement remains manual, is very time-consuming, inevitably contains some subjectivity, and the use of classes is limited to capture the continuous nature of snow types. Trying to overcome some of the two first limitations by automatic classification is of great interest. Different attempts exist to relate the SMP signal to scalar microstructural features of snow based on the physical interpretation of the penetration process (e.g., Löwe & van Herwijnen (2012), Lin et al. (2022)) or with direct statistical / machine learning approaches (e.g., Proksch et al. (2015)). In particular, King et al. (2020) and Satyawali et al. (2009) used the latter approach to relate MEASURED grain shape class to SMP profiles. Here the ground truth is not the measured grain shape on independent data but corresponds to the interpretation of solely the SMP signal signature.*

○ Thank you for your comment – it made very clear to us that we have not communicated the labeling process and how we obtained the "ground truth" data detailed enough. We are sorry for having left out this crucial part of our study in the manuscript.

○ Reasons why no measured grain shapes were used:

■ The conditions on Arctic sea ice are somewhat harsh and there were significant **time-constraints** involved during data collection. This means that only 1.5 hours were available to perform all measurements within one snow pit. It is not possible to collect all the in-situ observations one would like to have in such a setting. Manual snow stratigraphy analysis takes a

lot of time.

- **Temperature and wind** meant that looking at snow grains on a metal plate was almost impossible in the high-Arctic winter.

- Including the stratigraphy of snow is **very subjective**, i.e. it does vary from person to person. During the MOSAiC expedition, different people have been making measurements, which would lead to very different snow stratigraphy analysis and inconsistent / discontinuous profiles. Hence, this step was skipped. The SMP provides an analysis that will yield the same force profile no matter who measures it. We recognise that we introduce subjectivity at a later stage, however this is through one dataset trainer who has alternative measurements to hand to assist the classification, namely microCT and NIR. We therefore counteract the impossible classification in the field, with an alternative approach. A later study could work to compare these two methods, but that is beyond the scope of this paper, which primarily focuses on the development of ML-algorithms.

- Objective: In general, you do have the volume on sea ice, but not the time. As scientists, we have to deal with the situations at hand and it is still our mission to get the most out of the data that has been collected. Having an ML model that can help us process this kind of data (where we have volume but no time for measuring grain shapes) is exactly the objective of this paper.

- How the ground truth data was obtained / how the SMP signal interpretation was verified:

  - We are sorry that we have not pointed this out in the manuscript (the main author was not aware of the specific process): Micro-CT and NIR have been used in parallel to the SMP signals to fine-tune the classifications. The Micro-CT and NIR – where available – were used to confirm the shape of the grains.

- Further information we want to share:

  - On approximation: With the models presented in this study one is able to classify SMP profiles in a *simpler structure*. We are essentially doing the following: 1) Grouping signal types. 2) Identify from Micro-CT which grain types match the group the best. So, when we call one of those groups "depth hoar" it is clearly an approximation. All those labels are just approximations of the signal types. And we are using those labels because this is how we have established as a scientific community that

this is our common language to talk about snow.

- On uniqueness / the word "ground truth": We got rid of the word "ground truth" because it clearly eludes the fact that there is not such a thing as "ground truth". *Every* kind of classification of snow grain is always subjective. Whenever you put three experts in front of the same profile, you will get three slightly different classifications. This is not mitigated by snow pits.
  And also the ML models remain estimators. One classifies a number of profiles with a certain set of signals. This is now the "base data". Given this base data, we can now **estimate** for all the other thousands of profiles how they are stratified. Different models will also yield different segmentations and classifications which illustrates once more that a unique classification continues to elude us. There is no unique classification and there is no ground truth for this kind of data.

- How we will adapted the manuscript to clarify those things:

  - We will adapt the data section (we are adding two paragraphs about the data labeling).

  - We are adding a supplementary material laying out the details of the labeling procedure.

  - Removing the word "ground truth".

- *This direct identification has never been documented so far. The description in the text is elusive, with a reference (l.76, Schneebeli et al. 1999) that does not describe the procedure. Besides, the data presented here relies on the interpretation of a single expert (l. 75-77). One cannot evaluate any reproducibility of the procedure or agreement with ground truth based on manual observation in snow pit data. Moreover, it is highly likely that the estimation is subjective. For instance, in Fig. 1, one may wonder why only the specific layer at a depth between 98 and 102 mm is labeled as « Depth hoar wind packed » and not other layers below that show similar features. In addition, there are obviously « inconsistencies in their ground truth labeling » (l. 324) and the results are linked to « different classification styles of experts » (l.332) and the evaluation is qualitative (« classification patterns […] were satisfying to domain experts » l.368). The discussion is not convincing based only on the feeling that «in the view of the authors, a temporally consistent classification is more relevant to the interpretation of the development of the snowpack, even if there is a certain, but unknown, bias to an expert interpretation » (l. 255-257). To me, it appears, in the end, that the presented algorithms are able to reproduce one analysis of one single expert on specific snowpack types. In my opinion, this limits a lot of the interest and generalization to the snow community.*

○ Thank you for pointing out that the Schneebeli et al. (1999) reference is not providing fully the necessary information. We reference Schneebeli et al. (1991) here because they have shown for the first time that depth hoar and rounded grains can be visually separated in the SMP signal. Thus we base our work on the idea described there. However, it is true that the direct identification has not been documented in detail so far. Consequently, we decided to describe this process now in our supplementary material. We find that this documentation is not the main objective of this work but we definitely agree that the reader must have access to how this process works and how it was specifically conducted within this study.

○ We would like to point out that this paper submitted to the Geoscientific Model Development journal is really a *methodological* paper, analyzing which ML algorithms can help to automate SMP profile classification. It is not the objective of the paper to "classify snow according to the international guidelines with the SMP". Our goal is to classify a huge number of SMP profiles and get their layer segmentation by grouping signals. We are doing this for convenience, to make the life of SMP users easier. To summarize: This is a methodological paper and we are not deriving any conclusions from the data we used (we could use any dataset) – it is really about the methods.

○ It is true that the models provided here cannot be applied to any snowpack in the world. The algorithms and the framework presented can be applied to any snowpack – but one needs to retrain the algorithm. It may be the topic of another paper to provide a model that has been trained on a generalized dataset. This future study might profit from our work here, because they might be interested in learning which ML models are particularly suitable for this task. They can then use our repository – or extend it even further with other models if desired – and train it on a general dataset. Before being able to do so, we think it is valuable to have a fair comparison between a set of algorithms and an established model that can be re-trained to solve this task on any kind of SMP dataset.

○ Regarding the subjectivity of the classification: Yes, we agree. Machine learning works like this: Subjective data in – subjective classification out. Objective data in – objective classification out. This is why clean, unbiased, diverse datasets are so crucial in the machine learning domain. For the task described in this paper: Yes, you will absolutely get a subjective classification if you train your model on subjective labeling. If you can provide the models with a training dataset that is – in your view – objective, where snow experts have been collaborating and the complete community agrees all together that those are the right labels – then you can train the models and reproduce the labeling process of this communal labeling. Right now, we still face the following reality: If you ask for a snow grain classification in the field, then you ask for a classification from *one* person. This person will provide you with a subjective classification to date. We do think that

our field is still at a point where such a subjective classification is our reality and we cannot provide a general alternative to this within the study presented here. We are mainly making the life of this one person easier – we are upscaling the (subjective) classification process. This is the main objective of this study.

- ○ Changes that will be made to address the concerns:

    - ■ We will explain the context of the Schneebeli et al. (1999) citation.

    - ■ We will add an appendix where the labeling process is described in detail.

    - ■ We will add a short summary in the labeling paragraph to give an overview of the labeling process.

- *The authors refer to Fierz et al. (2009) for describing the different snow types referenced in the paper (l. 74). However, it is not very clear how the different classes presented in the paper (see legend of Fig. 1) are defined as they are not present in the international classification described in Fierz et al. (2009)*

    - ○ We are now clarifying further in the data section that we are merely "leaning" towards Fierz et al. (2009). We had to extend the classification because we worked with snow on Arctic sea ice – and the guidelines were not developed for snow on sea ice. The international classification is dominated by Alpine snow experts. The different types of depth hoar are typical for Arctic sea ice and not included in Fierz et al. (2009). We seek a language that is common to the snow community, but because of the specific region, we are not fully adhering to the international classification guideline and extend it where necessary.

    - ○ Changes: We will add a paragraph in the data section to clarify this.

- *Grain shape class has been used since the beginning of snow science and was first motivated by avalanche forecasting. It remains the most common descriptor in snowpit observations but has many known limitations. It is a discrete class whose evolution cannot be described by differential equations in models. It cannot be quickly and objectively described. Currently, the international classification is not necessarily adapted to any snow on Earth (e.g., here, the authors added classes that are not in the classification). Therefore, one may wonder why, in general, we want to stick to this description of snow.*

    - ○ Grain shape is also used within the polar community. The international classification – as pointed out by you – is also in our view just an approximation, as every classification is. We think we should stick to the language that is common to the community and extend it where necessary.

- ○ Changes: We will add a sentence that motivates why we are using Fierz et al. (2009) at all (common language).

- *The interest of the algorithm is described in grandiose terms: they make « training of interdisciplinary scientists in snow type categorization obsolete » (l. 31), «can be directly employed by practitioners for their own SMP datasets in the field » (l. 250), « These findings will enable SMP practitioners to automatically analyze their SMP measurements. To that end, an SMP user must simply decide on one of the fourteen models provided » (l. 369-370). However, I do not understand these sentences. I understood that everything relies on a single expert analysis, that the model must be retrained on other data (e.g., snow data in other places around the world) and that without this expert, no model can be retrained. In contrast, the limits of previous studies are somehow presented unfairly. For instance, it is indicated that « This [generalization] would not have been possible in previous works such as Satyawali et al. (2009) since knowledge rules for one snow region and season do not transfer to other regions or seasons » (l. 335), but the exact same applies to their work as the model must be retrained in any case to be used on other snowpack climate or expert analysis in the end (the model of Satyawali et al. (2009) could be retrained too).*

  - ○ Thank you for telling us that the applicability of the method did not become clear for you.

  - ○ Grandiose terms: We adapted that regarding the training of the interdisciplinary scientists, we understand that this can be misunderstood. We still think that it can be directly employed by practitioners in the field for their own SMP datasets, as we will try to point out in the following. One does not need to be an expert for that – the model can be retrained without the expert.

    - ■ Situation A:
      Situation: You are an expert, you collected a large dataset of SMP profiles. You are okay with manually labeling ~100 profiles – you might even have additional in-situ observations for those profiles and can manually label them. However, you do not want to do this manually for all profiles and have maybe hundreds or thousands of other profiles where no in-situ observations exist. You do know though, that they experienced similar snow conditions to the ones where you have in-situ observations.
      How would you use snowdragon:
      You would git clone the repository. You would choose a model to your liking. You would train the model on your dataset with the suggested parameters (or even load the weights we have used). If desired, you can tune the model (with the scripts provided in the repository). Afterwards, you can produce predictions for the unlabelled SMP profiles. This enables you to perform a quantitative analysis on your complete SMP dataset, i.e. you can estimate the occurrences of certain grain types, etc. Not every

single profile will be a perfectly classified profile, but you will be able to make meaningful estimations and quantitative analyses on your dataset.
Why is this better than using previous work:
*Note:* Feel free to use any of their methods – we have included random forest, nearest neighbors, and SVMs as well in snowdragon. This will enable you to compare those methods with each other.
*Random forest (original):* You would need to have the patience to segment all (every single of your thousand profiles) your profiles before you can classify anything. And you need to accept that you can only have layer analysis for layers > 20 mm.
*Nearest-neighbor (original):* You might first struggle with replicating their work and will then hand-craft specific knowledge rules to improve the performance of nearest neighbors. Those rules might not be straightforward since they are specifically crafted to circumvent errors that nearest neighbors consistently produce on your specific dataset.
*SVM (original)*: You have to ensure that each profile can be mapped to a specific density cutter measurement. If you do not have such a measurement directly related to your SMP profile (e.g. because it was cold and you could not make dozens of snow pits in a short amount of time but you still wanted to capture snow stratigraphy spaced out), you will not be able to use that method.
*Endnote:* All of the original methods work well – on their specific dataset with the circumstances given there. This means that there are a lot of strings attached: you need to provide additional snowpit information, manually segment data, and be happy with just three grain types or craft knowledge rules to push the performance of a method. We essentially try to provide a method with no strings attached and that is easier to re-use for your specific situation than all the previous ones. In contrast to previous work, one can also actually compare all those methods with each other in a fair comparison which has not been possible before.

- Situation B:
  Situation: You are not an expert in snow stratigraphy. Let's say you are a remote sensing specialist. You would like to get "ground-truth" data for a snow remote sensing project and you are provided with SMP profiles from the ground. In the best case you have an expert who can label some of your profiles. If you do not have that, you can search for a labeled dataset from the same region. (Of course it is helpful to communicate with an expert on this! The expert is not made completely obsolete – please pardon this exaggeration).
  How would you use snowdragon:
  Same procedure as explained above. You can relate the SMP classifications to your remote sensing data and can thus make snow-stratigraphy estimations on a large scale.

Why is this better than using previous work:
In the case of random forest and SVM you have to find snow experts who are willing to either manually segment all your profiles or you have to take a snow expert with you to make density cutter measurements. Both of these take a considerable amount of time and commitment. Finding a suitable labeled dataset or asking a snow expert for a few labeled profiles (possibly fine-tuned with in-situ observations) is much more realistic. The nearest neighbor approach suffers from the hard-coded knowledge-rules and that you can have no mixed layers, as explained above. The rules need to be adapted for the particular data and in contrast to labeled datasets, they are not simply publicly available but need to be carefully crafted and adapted. Furthermore, we can hope that snowdragon will be trained on a more generalized SMP dataset one day, meaning there is the possibility that the labeling step is not necessary anymore at a certain point. We do not have this possibility for the previous work – constructing knowledge rules, manual segmentation, and density cutter measurements cannot be skipped because they are inherent parts of those methods. These things are not *transferable.* We are providing a method where knowledge – in the form of labeled SMP datasets – can be transferred to other datasets. This means, if you are working on alpine snow, you can train your model on publicly available alpine snow SMP datasets right now. This is something that has not been possible before.

- General comment:
  We do understand that a Meta-Snowdragon model would have a much larger impact on the scientific community, since people would not need to retrain the model and might be able to apply the models on any snowprofile. However, the work we present here has the objective to show that ML algorithms can be used for SMP classification in practice – there is still work that needs to be done to make it easily usable for everyone without requiring people to put any additional (labeling) work into this. Also, a word of warning: A Meta-model that can be used for any snow profile might do well on average, but will perform worse on very specific/special datasets (like snow on Arctic sea ice) than a fine-tuned model. One could provide a Meta-model that can then be fine-tuned to specific datasets when necessary. We invite the community to take our work further and provide such a Meta-model. We suggest investigating transformer models to this end.

- Changes:

  - The sentence "obsolete" will be adapted to less grandiose terms.

- ■ We will add a user guide to make the process of how snowdragon can be applied more transparently.

- ● *The authors positively present the work as both « automatic classification and segmentation » (title and in the text) of snow profiles. It appears that no segmentation procedure is present in the paper. Indeed, the segmentation consists of saying that connected (i.e., neighboring) points with the same label belong to the same segment.*

  - ○ Exactly, our segmentation procedure consists of joining neighboring points that share the same class together into one segment. This is arguably a very naive segmentation method, but we wanted to keep it simple and focus the comparison study on the classification models. Future work could compare in more detail different segmentation methods or – as already suggested – look into first-segment-then-classify approaches. This may be a greater challenge from the classification model perspective since you have to deal with time series of different lengths and classify those accordingly. This is in general regarded as a somewhat greater challenge for machine learning models and we would be very curious to learn how those models are performing on this task.

  - ○ Regarding your question about why we present this work as both automatic classification and segmentation – despite the naive approach of the segmentation: Since e.g. Havens et al. (2012) really *only* classify profiles, i.e. the profiles have to be *manually segmented* first, it was important to us to make clear that our work does not require a manual segmentation of the SMP profiles.

  - ○ Suggested changes:

    - ■ Changing the title of the paper to:
      "Automatic classification of Snow Micro Penetrometer profiles with machine learning algorithms"
    - ■ We will go through the text and change the wording where we find it appropriate.

**The form of the work**

"On the form, the description of the work is sometimes vague and incomplete"

- ● *The objective of the paper described in l23-31 seems rather unclear to me. It took me several reads to understand that the goal is to reproduce the classification of one expert on SMP data.*

  - ○ Thank you for pointing this out. We will adapt the paragraph in the following way:

    "Traditionally, snow stratigraphy measurements are made in snow pits. These

pits are dug manually, and vertically into snowpacks and require trained operators and a substantial time commitment. To accelerate these measurements, the SnowMicroPen (SMP), a portable high-resolution snow penetrometer, can be used (Johnson and Schneebeli, 1998). Schneebeli and Johnson (1998) have demonstrated the SMP as a capable tool for rapid snow type classification and layer segmentation. The measurement results are stored in an SMP profile that consists of the penetration force signal of the measurement tip in Newton and the depth signal indicating how far the tip moved. Afterward, the SMP profiles must be manually labeled by an expert, which requires time, practice, and becomes infeasible for large datasets.

To address these shortcomings, Machine learning (ML) algorithms could be used to automate the labeling process. Instead of manually labeling each SMP profile, an ML model can be trained on a few labeled profiles and can subsequently reproduce the labeling patterns on other profiles. As a consequence this would (1) immensely accelerate the SMP analysis, (2) enable the analysis of large datasets, and (3) support interdisciplinary scientists who are unfamiliar with snow type categorization."

- *There is a welcome short bibliography on previous attempts to classify SMP profiles automatically. The description of the selected articles (Satyawali et al.,(2009), Havens et al., (2012), King et al., (2020)) would benefit from more detailed statements to capture what was really done in these papers. For instance, what is « too small to be representative » (of what?) (l. 34), « including knowledge-based rules » (l. 35), « good accuracy » (l. 42), and « additional snowpit information » (l. 42)? «*

  - Thank you very much for pointing out this unclarity. We have indeed cut the text a bit too drastically at this point. We will extend the literature review, mentioning now what "too small to be representative" means, "knowledge-based rules", "good accuracy", and specifying what the "additional snowpit information" is.
  - Adapted paragraph:

    "Several previous works have addressed the task of automatically classifying snow grain types with machine learning algorithms. The nearest neighbor method of \citet{satyawali2009preliminary} was the first model that automated both segmentation and classification of SMP profiles without needing additional snow pit information. To assign a grain type to an unlabelled data point, the method chooses the most frequent class occurring in the neighborhood of this data point. The neighborhood are those labeled data points that are most similar to the unlabeled data point. Their algorithm predicts five different snow types (``New Snow'', ``Faceted Snow'', ``Depth Hoar'', ``Rounded Grains'', ``Melt-Freeze''), with accuracies ranging from 0.68 to 0.94. However, this high performance is only achieved by integrating specific and inflexible expert rules. For example, one rule ensures that no ``Faceted Snow'', ``Depth Hoar'', or ``Rounded Grains'' occur

between layers of ``New Snow'', but exactly this happens under certain circumstances as they point out themselves. Hard-coded rules might improve the performance on one dataset, but they cannot capture all phenomena and will not generalize well to other datasets. The performance results are also limited by the fact that their testing set consists of only three SMP profiles, i.e. it is not clear how representative their results really are. In addition, their results can hardly transfer to the real-world setting because they explicitly exclude any mixed grain type layers. If an automatic segmentation and classification algorithm is intended to work with profiles straight from the field, this algorithm should be able to handle mixture classes, diverse snow phenomena, and be thoroughly tested.

\citet{havens2012automatic} worked with random forests and SVMs to classify SMP profiles. They used previously segmented SMP profiles and classified the grain type of each layer with the help of a random forest model. Their work builds upon their previous work with single decision trees \citep{havens2010singleCT}. They trained the model on three different grain types (``New Snow'', ``Rounded Grains'', ``Faceted Grains''), achieving error rates between 16.4\% and 44.4\% (depending on the dataset). Notably, \cite{havens2012automatic} requires profiles that have been manually segmented beforehand. Since this is done manually, this takes a considerable amount of time, raising the question to what extent the task has really been ``automated''. Moreover, only layers larger than 100 mm (sometimes 20 mm) could be considered due to the manual segmentation. In the field, particularly for avalanche risk assessment \citep{lutz2007segmentation_moving_window}, it is important to detect layers of only a few millimeter thickness as well. Improving on the work of \citet{havens2010singleCT} would thus include more grain types, thinner layers, and no need for manual segmentation.

More recently, \citet{king2020local} trained Support Vector Machines (SVMs) on SMP force signals and manual density cutter measurement. Both segmentation and the classification is conducted automatically. They distinguish three types of snow grains (``Rounded'', ``Faceted'' and ``Hoar'') and achieve classification accuracies between 0.76 and 0.83. The profiles were collected on Arctic ice around the same location, which means that the profiles might be more homogeneous than in other datasets. The model's generalisability could in theory be enhanced by training it on additional, broader datasets. Most importantly, the SVM method by \citet{king2020local} relies on additional manual density cutter measurement, time-intensive snow pit measurements that are not always available. Thus, similarly as for \citet{havens2012automatic}, more snow grain types would make the work more applicable in the field, as well as eliminating the necessity of additional manual density cutter measurements.
In summary, previous work showed that supervised machine learning algorithms are a promising pathway to automatic snow grain categorization."

- *Fig. 1, the international classification (Fierz et al.,2009) provides a color code. Is there a specific reason for not using it?*

  - 1) Since we only "lean to the classification" as discussed above (not all grain types present there, etc.) we had to add other colors.

  - 2) As stated by Fierz et al. (2009) themselves: "The colour convention is not optimized for people affected by colour vision deficiencies."

- *One key piece of information about the procedure is the list of predictors used as input for the ML model. They are very shortly described l. 79-86. But the description is too elusive to understand which variables are used. What are « added additional features», « time-dependent information » (where is time here ???), and « including variables of the shot noise model » (which variables?)?*

  - Time: Snow accumulated through time / SMP tip measuring the penetration force one time step after another. We adapted the word since time-dependent information seems to be a jargon that is more common in the ML community. We call it now "depth-dependent" information throughout the manuscript.

  - Features: We will adapt the paragraph and specify which features are included. We will also add a table in the appendix that lists all the features and provides an explanation for each feature.

  - Adapted paragraph:
    "For each SMP profile, we replaced negative force values with 0, summarized the signal into bins (1 mm), and added mean, variance, maximum, and minimum force values for those bins. Those values were also determined for a 4 mm and 12 mm moving window. Moreover, \citet{lowe2012poisson}' Poisson shot noise was used to extract $\delta$, $f$, $L$, and the median force value for a 4 and 12 mm window. We added further depth-dependent information by including for each data point the distance from the ground and position within the snowpack. Refer to Table \ref{tab:features} in Appendix \ref{app:features} for an overview of all features used for each SMP profile, and to Table \ref{tab:feature_corr} to see the feature importance for each grain type."

---

## Referee Report (RR1)

**egusphere-2022-938 referee report 3**

**Detailed comments**

Please note that all line numbers refer to the ATC2 document where changes from the previous manuscript version are highlighted.

- *title*: Classification is a broad term. Besides grain type classification, there could also be other types of classification, e.g. by stability of the snowpack for avalanche hazard contexts. I therefore suggest including the focus on grain types. E.g., "Automatic grain type classification of Snow Micro Penetrometer profiles with machine learning models".

- L22–24: « This knowledge helps to discern fundamental snow and climate mechanisms in the Arctic and to analyze polar tipping points. Classification of snow types is essential to assess the state of our cryosphere and is thus of interest for polar, cryospheric, and climate change research. »

    i) Please include references here.

    ii) In your response to referee 2 you nicely explain why knowledge about grain type profiles is relevant for remote sensing in an arctic context. Please include one or two sentences that explicitly state the need for classifying grain types and possible implications thereof.

    iii) I am more familiar with the terms "grain type" or "snow grain type" than "snow type". In the newly added paragraphs starting at L42ff you also primarily use the term "grain type". Please use the full term "snow grain type" here in the beginning to make its meaning explicit. In the following I suggest you use one term consistently, but I leave this up to you.

- L117: « The labels indicate.. ». Please state more explicitly., e.g., "The labels expressed by color in Fig. 1 ..."

- Footnote 1: Fierz et al "refer" (without s)

- Figure 1:

    i) The background color in the inset panel refers to the manual human labels. If the colors in the main panel also refer to that manual labelling, please state explicitly. Please change the wording « ground truth » in the caption.

    ii) I understand the discussion around expert labelling, its advantages and limitations, and I fully support the presented approach in the context of this methodological contribution and demo application (one expert who did the labelling partially with additional in-situ measurements, and one person reviewing it). However, I have one comment about the specific labels in the inset panel that seem inconsistent to me: The surface layer is labelled as « Rounded Grains Wind Packed » with a mean force around ~2N. The spike of ~6N at 100mm depth is labelled as « Depth Hoar Wind Packed ». I am neither an expert in the interpretation of SMP signals nor an expert of Arctic snow conditions, but it seems odd to me that DHwp would be three times harder than RGwp. Table B1 states that RGwp ranges between 10–40N.
    I don't think that this is a big deal, and I assume that you have good reasons for your choices. Nevertheless, potential users of your tools might be situated in Alpine conditions and get similarly confused as I did. I suggest you extend your discussion of the labelling process in Appendix B with the most important particularities encountered in your data set and environment conditions. This should also include a qualitative description of the grain types that are *not* included in the International Classification For Seasonal Snow On The Ground (e.g., DHwp, ...). You can also consider picking an SMP signal for Figure 1 that demonstrates unambiguous (textbook-like) labels.
    Please see this comment also in light of my general comment earlier. The fact that all three referees spent considerable amount of thinking around the subjectivity of the manual labels and the resulting impact on the predictions highlights the potential benefit of a more in-depth approach (/visualization) to look at the data set.

- L113–146: The flow of the paragraphs and their storyline jumps back and forth a little bit. I suggest to reorder the sequence of the paragraphs. e.g., (1) Intro to MOSAiC (incl. measurements and conditions), (2) explicit statement about the data used for this study, (2a) SMP, (2b) Micro-CT: please explain and cite, mention

full term before using acronym, state how many Micro-CTs were available, (2c) NIR: please explain and cite, mention full term before using acronym, state how many photographs were available.

- L153: Poisson shot noise *model*

- Figure 2: Please change the label « Ground Truth » in panel (a).

- L536: Can you cite these data sets?

- Appendix A: I appreciate this detailed user guide very much. Consider refering readers to that appendix when introducing the high-level process of labelling in the main body.

- Appendix B: Please include at least a brief explanation of how the Micro-CT and NIR measurements were used to inform/nudge/validate the grain type labelling of the SMP signal. I personally would appreciate one example figure that shows how an additional in-situ measurement made a challenging SMP signal more interpretable. A good example could be the signal in Fig. 3 that raised a concern with referee 1.

    – See related comment about Appendix B that I raised earlier (when writing about Fig 1)

- Figures Appendix B: I appreciate the figures that demonstrate typical SMP signals for different grain types very much. It would help a lot to see those signals in subplots in one figure on one page, rather than spread out. To make it even easier for the reader, you could clip the x-axis to the window that shows the desired signal and not show the remainder of the measurement. At least a background shading that highlights the window of interest would be nice.

- Appendix D: I apreciate the definitions of variable names, features, and metrics. Please reference these definitions in the main body of the manuscript when new terms come up.

- Figure G1: Please cross-reference the list in Appendix D that explains the variable names. When I jumped to the decision tree figure from the main body of the text I had skipped Appendix D, and was missing exactly this piece of information.

- One last question out of curiosity: What inspired the name « snowdragon » ;-)?

---

## Author Response (AR2)

**Authors' Response to RC3**

**General Response**

We thank the reviewer for their detailed feedback and think that their comments improved the manuscript further. We implemented all suggestions as you can see in the point-by-point response.

In response to the issues of double-checking the data and the subjectivity inherent in the manual labeling: We are glad that the reviewer found the new manuscript is more transparent on that matter. As requested, we added further information on the additional NIR and microCT measurements to make clearer how the "double-checking" during labeling worked.

Regarding the visualizations: Figure 6) shows predictions on unlabelled data and is not intended to help with comparing manual labels to ML predictions (since we need the manual labels for that). In order to visually compare ML predictions and manual labels we intended figures similar to Figure 3. Those figures are produced for every single profile and each model and are accessible on the GitHub repository alongside with further types of visualizations that have not been included in the paper. We have not fully understood which type of visualization is requested by the spaghetti/ridge plots, but would be happy to extend the GitHub repository with this type of visualization if given more detail.

We edited the manuscript to improve the paragraph structure, flow, and language use as suggested by the reviewer. We especially find the suggestion helpful to include a visualization of NIR, microCT and SMP measurements alongside and the suggestion on how to rewrite the paragraphs for the data section.

We appreciate the time and thought the reviewer has put into their feedback!

**Point-by-Point Response**

- *Title: Classification is a broad term. Besides grain type classification, there could also be other types of classification, e.g. by stability of the snowpack for avalanche hazard contexts. I therefore suggest including the focus on grain types. E.g., "Automatic grain type classification of Snow Micro Penetrometer profiles with machine learning models".*
    - We agree that classification is too broad of a term and we have adapted the title accordingly. Instead of calling it "grain type classification", we would like to call it "snow type classification" to follow the terminology used by the international classification for seasonal snow on the ground (Fierz et al, 2009).

- *L22–24: « This knowledge helps to discern fundamental snow and climate mechanisms in the Arctic and to analyze polar tipping points. Classification of snow types is essential to assess the state of our cryosphere and is thus of interest for polar, cryospheric, and climate change research. »*

  *i) Please include references here.*
    - Thank you for pointing this out, we included additional references in the revised manuscript.

  *ii) In your response to referee 2 you nicely explain why knowledge about grain type profiles is relevant for remote sensing in an arctic context. Please include one or two sentences that explicitly state the need for classifying grain types and possible implications thereof.*
    - We added the following sentences: "Grain type is often better reproduced in detailed snow cover models Vionnet et al., 2012) than their effective physical properties, especially indirectly structural anisotropy (King et al., 2015). This is especially relevant for active and passive microwave sensing, essential to map the arctic snowpack during polar night (Sandells et al., 2023)."

  *iii) I am more familiar with the terms "grain type" or "snow grain type" than "snow type". In the newly added paragraphs starting at L42ff you also primarily use the term "grain type". Please use the full term "snow grain type" here in the beginning to make its meaning explicit. In the following I suggest you use one term consistently, but I leave this up to you.*
    - We completely agree that it is more reader-friendly to use one term consistently. We were not sure which of the terms to use and we settled in the end with "snow type" since this is the term used by the by the international classification for seasonal snow on the ground (Fierz et al, 2009). We added a footnote after using the term for the first time in the introduction, listing different forms of the term.

- *L117: « The labels indicate.. ». Please state more explicitly., e.g., "The labels expressed by color in Fig. 1 . . . "*

- ○ Accepted.

- *Footnote 1: Fierz et al "refer" (without s)*
  - ○ Accepted.

- Figure 1:
  *i) The background color in the inset panel refers to the manual human labels. If the colors in the main panel also refer to that manual labelling, please state explicitly. Please change the wording « ground truth » in the caption.*
  - ○ Thank you for bringing this up - accepted.

  *ii) I understand the discussion around expert labelling, its advantages and limitations, and I fully support the presented approach in the context of this methodological contribution and demo application (one expert who did the labelling partially with additional in-situ measurements, and one person reviewing it).*
  - ○ We also think that it is important to have this discussion and we are glad to hear that you support the presented approach in this context!

  *However, I have one comment about the specific labels in the inset panel that seem inconsistent to me: The surface layer is labelled as « Rounded Grains Wind Packed » with a mean force around ~2N. The spike of ~6N at 100mm depth is labelled as « Depth Hoar Wind Packed ». I am neither an expert in the interpretation of SMP signals nor an expert of Arctic snow conditions, but it seems odd to me that DHwp would be three times harder than RGwp.*
  - ○ This is among the expected range for us: DHwp falls into the type of "hard depth hoar", described in Pfeffer et al. (2002). We added some sentences about that in Appendix B to provide the necessary information to the reader for that point.

  *Table B1 states that RGwp ranges between 10–40N. I don't think that this is a big deal, and I assume that you have good reasons for your choices. Nevertheless, potential users of your tools might be situated in Alpine conditions and get similarly confused as I did. I suggest you extend your discussion of the labelling process in Appendix B with the most important particularities encountered in your data set and environment conditions. This should also include a qualitative description of the grain types that are not included in the International Classification For Seasonal Snow On The Ground (e.g., DHwp, . . . ).*
  - ○ The range is indeed too high, we corrected this to 2-20 N.
  - ○ Since an in-depth description of the environmental conditions and the particularities encountered during the data collection is beyond the scope of this paper, we refer the reader to the following publication instead: https://online.ucpress.edu/elementa/article/9/1/00023/118092/Meteorological-conditions-during-the-MOSAiC
  - ○ We added additional grain type descriptions, however, only dhwp and mfdh seemed to need additional explanations since all other grain types are included in the international classification. We are happy to extend this section further if

requested.

*You can also consider picking an SMP signal for Figure 1 that demonstrates unambiguous (textbook-like) labels. Please see this comment also in light of my general comment earlier. The fact that all three referees spent considerable amount of thinking around the subjectivity of the manual labels and the resulting impact on the predictions highlights the potential benefit of a more in-depth approach (/visualization) to look at the data set.*

- ○ We hope that the extension of Appendix B addresses your comments. The SMP signal in the inset panel serves only the purpose of showing how the SMP signal looks like in general and should inform the reader about the general task at hand. The inset panel is not supposed to demonstrate how the signal of specific snow types look like (textbook-like). For that, we prefer to forward the reader to the Appendix B where the signals are visualized for each snow type. There, the signals are more visible than in the inset panel of Figure 1, plus we can show the whole range of signals and not just a subset.

- *L113–146: The flow of the paragraphs and their storyline jumps back and forth a little bit. I suggest reordering the sequence of the paragraphs. e.g., (1) Intro to MOSAiC (incl. measurements and conditions), (2) explicit statement about the data used for this study, (2a) SMP, (2b) Micro-CT: please explain and cite, mention 1full term before using acronym, state how many Micro-CTs were available, (2c) NIR: please explain and cite, mention full term before using acronym, state how many photographs were available.*
  - ○ Thank you, we agree that the flow can be improved. We restructured the content as suggested and added the information you requested. We added further information, especially on the numbers, citations, and explanations on NIR and micro-CT.

- *L153: Poisson shot noise model*
  - ○ Accepted.

- *Figure 2: Please change the label « Ground Truth » in panel (a).*
  - ○ Accepted.

- *L536: Can you cite these data sets?*
  - ○ We added three examples of publicly available SMP datasets.

- *Appendix A: I appreciate this detailed user guide very much. Consider refering readers to that appendix when introducing the high-level process of labelling in the main body.*
  - ○ Thank you - we have added a reference at the end of the introduction when outlining the rest of the paper. We thought that might be the best place (grouped together with data & code availability) since the user guide includes so many different aspects (collection, labeling, model selection, etc.).

- *Appendix B: Please include at least a brief explanation of how the Micro-CT and NIR measurements were used to inform/nudge/validate the grain type labelling of the SMP signal. I personally would appreciate one example figure that shows how an additional in-situ measurement made a challenging SMP signal more interpretable. A good example could be the signal in Fig. 3 that raised a concern with referee 1. – See related comment about Appendix B that I raised earlier (when writing about Fig 1).*
  - Thank you for that suggestion. Unfortunately, the requested SMP profile does not have any corresponding microCT or NIR images. We choose a set of SMP profiles instead for which we retrieved microCT and NIR images. Please take a look at Figure B2 that we included upon your suggestion. The new subsection of the appendix intends to give the reader a general intuition of how the different measurements are used in parallel, however, an in-depth explanation is beyond the scope of this paper. For further information, we refer the reader to the metadata section in the corresponding dataset publication (https://doi.pangaea.de/10.1594/PANGAEA.935934).

[Figure]

- *Figures Appendix B: I appreciate the figures that demonstrate typical SMP signals for different grain types very much. It would help a lot to see those signals in subplots in one figure on one page, rather than spread out. To make it even easier for the reader, you could clip the x-axis to the window that shows the desired signal and not show the*

*remainder of the measurement. At least a background shading that highlights the*
*window of interest would be nice.*
- ○ We summarized the figures into one figure with 8 panels spread across two
    pages. We hope that this gives a better overview. We highlighted the windows of
    interest with a thin red box. When there is no red box it means that the complete
    signal represent the mentioned snow type.

- *Appendix D: I appreciate the definitions of variable names, features, and metrics. Please*
  *reference these definitions in the main body of the manuscript when new terms come up.*
    - ○ The appendix about the metrics is referenced when the metrics are mentioned in
        the Section "Evaluation."
    - ○ The appendix about the features and variable names is referenced in the data
        section. We added the reference at several other locations where applicable.

- *Figure G1: Please cross-reference the list in Appendix D that explains the variable*
  *names. When I jumped to the decision tree figure from the main body of the text I had*
  *skipped Appendix D, and was missing exactly this piece of information.*
    - ○ We included this reference now.

- One last question out of curiosity: What inspired the name « snowdragon » ;-)?
    - ○ First author responding on her own: I am just a very big fan of dragons (they are
        my favorite fantasy animal). The snow-layer structure of a snowpack reminded
        me of the scales of a dragon - for some reason. And dragons are hoarding -
        since we implemented so many different ML models, I had the feeling we are
        building a dragon hoard of ML models at some point. As you can see it was
        purely association-driven :)

**Other Revisions**
- "Labeling" instead of "labelling", "labeled" instead of "labelled"
- Requested by copernicus:
    - ○ We adjusted the list of corresponding authors
    - ○ C2 and H1 are both inserted as figures and called  / referred to as "Tables",
        hence the request should already be satisfied?

---

## Author Response (AR3)

**Reply to minor revision**

Dear Dr Mausson

Thank you for your comments and request for a minor revison (in italics):

*"Please check one more time that all DOIs and links in the Code & Data availability section are up-to-date. You can also remove the "Additional notes" section - it was not my intent to suggest malpractice on your part, but maybe you could have cited this previous contribution in the paper somehow."*

We removed the "Additional notes" section, and cite the work in the introduction (line 80)

We checked the bibliography, and added DOIs to the references

Best regards

On behalf of all authors
Martin Schneebeli